# Provable Compositional Generalization for Object-Centric Learning

**Thaddäus Wiedemer**[1,2,3*]  **Jack Brady**[1,2,3*]  **Alexander Panfilov**[1,2,3*]  **Attila Juhos**[1,2,3*]
**Matthias Bethge**[1,3]  **Wieland Brendel**[2,3,4]

[1]University of Tübingen    [2]Max Planck Institute for Intelligent Systems
[3]Tübingen AI Center    [4]ELLIS Institute Tübingen

{thaddaeus.wiedemer, jack.brady, attila.juhos}@tuebingen.mpg.de

## ABSTRACT

Learning representations that generalize to novel compositions of known concepts is crucial for bridging the gap between human and machine perception. One prominent effort is learning object-centric representations, which are widely conjectured to enable compositional generalization. Yet, it remains unclear when this conjecture will be true, as a principled theoretical or empirical understanding of compositional generalization is lacking. In this work, we investigate when compositional generalization is guaranteed for object-centric representations through the lens of identifiability theory. We show that autoencoders that satisfy structural assumptions on the decoder and enforce encoder-decoder consistency will learn object-centric representations that provably generalize compositionally. We validate our theoretical result and highlight the practical relevance of our assumptions through experiments on synthetic image data.

## 1 INTRODUCTION

Despite tremendous advances in machine learning, a large gap still exists between humans and machines in terms of learning efficiency and generalization (Tenenbaum et al., 2011; Behrens et al., 2018; Schölkopf et al., 2021). A key reason for this is thought to be that machines lack the ability to *generalize compositionally*, which humans heavily rely on (Fodor and Pylyshyn, 1988; Lake et al., 2017; Battaglia et al., 2018; Goyal and Bengio, 2022; Greff et al., 2020). Namely, humans are able to recompose previously learned knowledge to generalize to never-before-seen situations.

Significant work has thus gone into the problem of learning representations that can generalize compositionally. One prominent effort is *object-centric representation learning* (Burgess et al., 2019; Greff et al., 2019; Locatello et al., 2020a; Lin et al., 2020; Singh et al., 2022; Elsayed et al., 2022; Seitzer et al., 2023), which aims to represent each object in an image via a distinct subset of the image's latent code. Due to this modular structure, object-centric representations are widely conjectured to enable compositional generalization (Battaglia et al., 2018; Kipf et al., 2020; Greff et al., 2020; Locatello et al., 2020a). Yet, it remains unclear when this conjecture is actually true because a theoretical understanding of compositional generalization for unsupervised object-centric representations is lacking, and empirical methods are frequently not scrutinized for their ability to generalize compositionally. Consequently, it is uncertain to what extent advancements in object-centric learning promote compositional generalization and what obstacles still need to be overcome.

In this work, we take a step towards addressing this point by investigating theoretically when compositional generalization is possible in object-centric representation learning. To do this, we formulate compositional generalization as a problem of *identifiability* under a latent variable model in which objects are described by subsets of latents called *slots* (see Fig. 1 left). Identifiability provides a rigorous framework to study representation learning, but previous results have only considered identifiability of latents *in-distribution* (ID), i.e., latents that generate the training distribution (Hyvärinen

---

*Equal contribution, order decided by dice roll.
    Code at github.com/brendel-group/objects-compositional-generalization

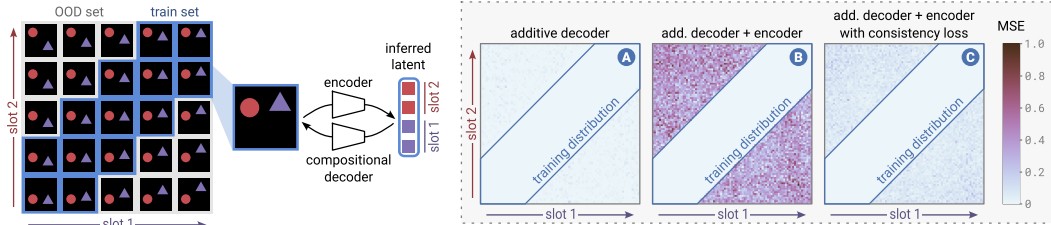

Figure 1: **Compositional generalization in object-centric learning**. We assume a latent variable model where objects in an image (here, a triangle and a circle) are described by latent *slots*. Our notion of *compositional generalization* requires a model to identify the ground-truth latent slots (*slot identifiability*, Def. 2) on the train distribution and to transfer this identifiability to out-of-distribution (OOD) combinations of slots (Def. 3). An autoencoder achieves slot identifiability on the train distribution if its decoder is *compositional* (Thm. 1). Further, we prove that decoders that are *additive* are able to generalize OOD as visualized in **(A)** via the isolated decoder reconstruction error over a 2D projection of the latent space (see App. B.3). However, this does not guarantee that the entire model generalizes OOD, as the encoder will generally not invert the decoder on OOD slot combinations, leading to a large overall reconstruction error **(B)**. To address this, we introduce a *compositional consistency* regularizer (Def. 6), which allows the full autoencoder to generalize OOD (**C**, Thm. 3).

et al., 2023). Compositional generalization, however, requires identifying the ground-truth latents not just ID, but also *out-of-distribution* (OOD) for unseen combinations of latent slots.

We pinpoint the core challenges in achieving this form of identifiability in Sec. 2 and show how they can be overcome theoretically by autoencoder models which satisfy two properties: *additivity* (Def. 5) and *compositional consistency* (Def. 6). Informally, additivity states that the latents are decoded as the sum of individual slot-wise decodings, while compositional consistency states that the encoder inverts the decoder ID as well as OOD. When coupled with previous identifiability results from Brady et al. (2023), we prove that autoencoders that satisfy these assumptions will learn object-centric representations which *provably generalize compositionally* (Thm. 3).

We discuss implementing additivity in practice and propose a regularizer that enforces compositional consistency by ensuring that the encoder inverts the decoder on novel combinations of ID latent slots (Sec. 4). We use this to empirically verify our theoretical results in Sec. 6.1 and find that additive autoencoders that minimize our proposed regularizer on a multi-object dataset are able to generalize compositionally. In Sec. 6.2, we study the importance of our theoretical assumptions for the popular object-centric model Slot Attention (Locatello et al., 2020a) on this dataset.

**Notation**    Vectors or vector-valued functions are denoted by bold letters. For vectors with factorized dimensionality (e.g., $z$ usually from $\mathbb{R}^{KM}$) or functions with factorized output (usually $\hat{g}$ mapping to $\mathbb{R}^{KM}$), indexing with $k$ denotes the $k$-th contiguous sub-vector (i.e., $z_k \in \mathbb{R}^M$ or $\hat{g}_k(x) \in \mathbb{R}^M$). Additionally, for a positive integer $K$ we write the set $\{1, \ldots, K\}$ as $[K]$.

## 2    PROBLEM SETUP

Informally, we say that a model generalizes compositionally if it yields an object-centric representation for images containing *unseen combinations* of *known objects*, i.e., objects observed during training (Zhao et al., 2022; Frady et al., 2023; Wiedemer et al., 2023). For example, a model trained on images containing a red square and others containing a blue triangle should generalize to images containing both objects simultaneously—even if this combination has not previously been observed.

To formalize this idea, we first define scenes of multiple objects through a latent variable model. Specifically, we assume that observations $x$ of multi-object scenes are generated from latent vectors $z$ by a *diffeomorphic generator* $f : \mathcal{Z} \rightarrow \mathcal{X}$ mapping from a *latent space* $\mathcal{Z}$ to a *data space* $\mathcal{X} \subseteq \mathbb{R}^N$, i.e., $x = f(z)$ (see also App. A.1). Each object in $x$ should be represented by a unique sub-vector $z_k$ in the latent vector $z$, which we refer to as a *slot*. Thus, we assume that the latent space $\mathcal{Z}$ factorizes into $K$ slots $\mathcal{Z}_k$ with $M$ dimensions each:

$$\mathcal{Z}_k \subseteq \mathbb{R}^M \quad \text{and} \quad \mathcal{Z} = \mathcal{Z}_1 \times \cdots \times \mathcal{Z}_K \subseteq \mathbb{R}^{KM}. \tag{1}$$

A slot $\mathcal{Z}_k$ contains all possible configurations for the $k$-th object, while $\mathcal{Z}$ encompasses all possible combinations of objects. For our notion of compositional generalization, a model should observe all possible configurations of each object but not necessarily all combinations of objects. This corresponds to observing samples generated from a subset $\mathcal{Z}^S$ of the latent space $\mathcal{Z}$, where $\mathcal{Z}^S$ contains all possible values for each slot. We formalize this subset below.

**Definition 1** (Slot-supported subset). For $\mathcal{Z}^S \subseteq \mathcal{Z} = \mathcal{Z}_1 \times \cdots \times \mathcal{Z}_K$, let $\mathcal{Z}_k^S := \{ \boldsymbol{z}_k | \boldsymbol{z} \in \mathcal{Z}^S \}$. $\mathcal{Z}^S$ is said to be a *slot-supported subset* of $\mathcal{Z}$ if $\mathcal{Z}_k^S = \mathcal{Z}_k$ for any $k \in [K]$.

One extreme example of a slot-supported subset $\mathcal{Z}^S$ is the trivial case $\mathcal{Z}^S = \mathcal{Z}$; another is a set containing the values for each slot *exactly once* such that $\mathcal{Z}^S$ resembles a 1D manifold in $\mathbb{R}^{KM}$.

We assume observations $\boldsymbol{x}$ from a *training space* $\mathcal{X}^S$ are generated by a slot-supported subset $\mathcal{Z}^S$, i.e., $\mathcal{X}^S := \boldsymbol{f}(\mathcal{Z}^S)$. The following generative process describes this:

$$\boldsymbol{x} = \boldsymbol{f}(\boldsymbol{z}), \quad \boldsymbol{z} \sim p_{\boldsymbol{z}}, \quad \mathrm{supp}(p_{\boldsymbol{z}}) = \mathcal{Z}^S. \tag{2}$$

Samples from such a generative process are visualized in Fig. 1 for a simple setting with two objects described only by their y-coordinate. We can see that the training space contains each possible configuration for the two objects but not all possible combinations of objects.

Now, assume we have an inference model which only observes data on $\mathcal{X}^S$ generated according to Eq. 2. In principle, this model could be any sufficiently expressive diffeomorphism; however, we will assume it to be an *autoencoder*, as is common in object-centric learning (Yuan et al., 2023). Namely, we assume the model consists of a pair of differentiable functions: an *encoder* $\hat{\boldsymbol{g}} : \mathbb{R}^N \to \mathbb{R}^{KM}$ and a *decoder* $\hat{\boldsymbol{f}} : \mathbb{R}^{KM} \to \mathbb{R}^N$, which induce the *inferred latent space* $\hat{\mathcal{Z}} := \hat{\boldsymbol{g}}(\mathcal{X})$ and the *reconstructed data space* $\hat{\mathcal{X}} := \hat{\boldsymbol{f}}(\hat{\mathcal{Z}})$. The functions are optimized to invert each other on $\mathcal{X}^S$ by minimizing the reconstruction objective

$$\mathcal{L}_{\mathrm{rec}}(\mathcal{X}^S) = \mathcal{L}_{\mathrm{rec}}(\hat{\boldsymbol{g}}, \hat{\boldsymbol{f}}, \mathcal{X}^S) := \mathbb{E}_{\boldsymbol{x} \sim p_{\boldsymbol{x}}} \left[ \left\| \hat{\boldsymbol{f}}(\hat{\boldsymbol{g}}(\boldsymbol{x})) - \boldsymbol{x} \right\|_2^2 \right], \quad \mathrm{supp}(p_{\boldsymbol{x}}) = \mathcal{X}^S. \tag{3}$$

We say that an autoencoder $(\hat{\boldsymbol{g}}, \hat{\boldsymbol{f}})$ produces an object-centric representation via $\hat{\boldsymbol{z}} := \hat{\boldsymbol{g}}(\boldsymbol{x})$ if each inferred latent slot $\hat{\boldsymbol{z}}_j$ encodes all information from exactly one ground-truth latent slot $\boldsymbol{z}_k$, i.e., the model separates objects in its latent representation. We refer to this notion as *slot identifiability*, which we formalize below, building upon Brady et al. (2023):

**Definition 2** (Slot identifiability). Let $\boldsymbol{f} : \mathcal{Z} \to \mathcal{X}$ be a diffeomorphism. Let $\mathcal{Z}^S$ be a slot-supported subset of $\mathcal{Z}$. An autoencoder $(\hat{\boldsymbol{g}}, \hat{\boldsymbol{f}})$ is said to *slot-identify* $\boldsymbol{z}$ on $\mathcal{Z}^S$ w.r.t. $\boldsymbol{f}$ via $\hat{\boldsymbol{z}} := \hat{\boldsymbol{g}}(\boldsymbol{f}(\boldsymbol{z}))$ if it minimizes $\mathcal{L}_{\mathrm{rec}}(\mathcal{X}^S)$ and there exists a permutation $\pi$ of $[K]$ and a set of diffeomorphisms $\boldsymbol{h}_k : \boldsymbol{z}_{\pi(k)} \mapsto \hat{\boldsymbol{z}}_k$ for any $k \in [K]$.

Intuitively, by assuming $(\hat{\boldsymbol{g}}, \hat{\boldsymbol{f}})$ minimizes $\mathcal{L}_{\mathrm{rec}}(\mathcal{X}^S)$, we know that on the training space $\mathcal{X}^S$, $\hat{\boldsymbol{g}}$ is a diffeomorphism with $\hat{\boldsymbol{f}}$ as its inverse. This ensures that $\hat{\boldsymbol{z}}$ preserves all information from ground-truth latent $\boldsymbol{z}$. Furthermore, requiring that the slots $\hat{\boldsymbol{z}}_k$ and $\boldsymbol{z}_{\pi(k)}$ are related by a diffeomorphism ensures that this information factorizes in the sense that each inferred slot contains *only* and *all* information from a corresponding ground-truth slot.[*] We can now formally define what it means for an autoencoder to generalize compositionally.

**Definition 3** (Compositional generalization). Let $\boldsymbol{f} : \mathcal{Z} \to \mathcal{X}$ be a diffeomorphism and $\mathcal{Z}^S$ be a slot-supported subset of $\mathcal{Z}$. An autoencoder $(\hat{\boldsymbol{g}}, \hat{\boldsymbol{f}})$ that slot-identifies $\boldsymbol{z}$ on $\mathcal{Z}^S$ w.r.t. $\boldsymbol{f}$ is said to *generalize compositionally w.r.t.* $\mathcal{Z}^S$, if it also slot-identifies $\boldsymbol{z}$ on $\mathcal{Z}$ w.r.t. $\boldsymbol{f}$.

This definition divides training an autoencoder that generalizes compositionally into two challenges.

**Challenge 1: Identifiability** Firstly, the model must slot-identify the ground-truth latents on the slot-supported subset $\mathcal{Z}^S$. Identifiability is generally difficult and is known to be impossible without further assumptions on the generative model (Hyvärinen and Pajunen, 1999; Locatello et al., 2019). The majority of previous identifiability results have addressed this by placing some form of statistical independence assumptions on the latent distribution $p_{\boldsymbol{z}}$ (Hyvärinen et al., 2023). In our setting, however, $p_{\boldsymbol{z}}$ is only supported on $\mathcal{Z}^S$, which can lead to extreme dependencies between latents (e.g.,

---

[*]Note that when $\mathcal{Z}^S = \mathcal{Z}$, we recover the definition of slot identifiability in Brady et al. (2023).

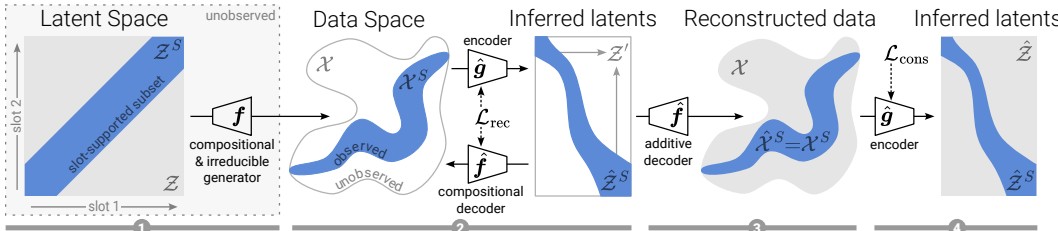

Figure 2: **Overview of our theoretical contribution**. (**1**) We assume access to data from a training space $\mathcal{X}^S \subseteq \mathcal{X}$, which is generated from a *slot-supported subset* $\mathcal{Z}^S$ of the latent space $\mathcal{Z}$ (Def. 1), via a *compositional* and *irreducible* generator $\boldsymbol{f}$. (**2**) We show that an autoencoder with a *compositional* decoder $\hat{\boldsymbol{f}}$ trained via the reconstruction objective $\mathcal{L}_{\text{rec}}$ on this data will *slot-identify* ground-truth latents $\boldsymbol{z}$ on $\mathcal{Z}^S$ (Thm. 1). Since the inferred latents $\hat{\boldsymbol{z}}$ slot-identify $\boldsymbol{z}$ ID on $\mathcal{Z}^S$, their slot-wise recombinations $\mathcal{Z}'$ slot-identify $\boldsymbol{z}$ OOD on $\mathcal{Z}$. However, the encoder $\hat{\boldsymbol{g}}$ is not guaranteed to infer OOD latents such that $\hat{\boldsymbol{g}}(\mathcal{X}) = \hat{\mathcal{Z}} = \mathcal{Z}'$. (**3**) On the other hand, if the decoder $\hat{\boldsymbol{f}}$ is additive, its reconstructions are guaranteed to generalize such that $\hat{\boldsymbol{f}}(\mathcal{Z}') = \mathcal{X}$ (Thm. 2). (**4**) Therefore, regularizing the encoder $\hat{\boldsymbol{g}}$ to invert $\hat{\boldsymbol{f}}$ using our proposed *compositional consistency* objective $\mathcal{L}_{\text{cons}}$ (Def. 6) enforces $\hat{\mathcal{Z}} = \mathcal{Z}'$, thus enabling the model to *generalize compositionally* (Thm. 3).

see Fig. 2 where slots are related almost linearly on $\mathcal{Z}^S$). It is thus much more natural to instead place assumptions on the generator $\boldsymbol{f}$ to sufficiently constrain the problem, in line with common practices in object-centric learning that typically assume a structured decoder (Yuan et al., 2023).

**Challenge 2: Generalization**  Even if we can train an autoencoder $(\hat{\boldsymbol{g}}, \hat{\boldsymbol{f}})$ that slot-identifies $\boldsymbol{z}$ *in-distribution* (ID) on the slot-supported subset $\mathcal{Z}^S$, we still require it to also slot-identify $\boldsymbol{z}$ *out-of-distribution* (OOD) on all of $\mathcal{Z}$. Empirically, multiple prior works have demonstrated in the context of disentanglement that this form of OOD generalization does not simply emerge for models that can identify the ground-truth latents ID (Montero et al., 2021; 2022; Schott et al., 2022). From a theoretical perspective, OOD generalization of this form implies that the behavior of the generator $\boldsymbol{f}$ on the full latent space $\mathcal{Z}$ is completely determined by its behavior on $\mathcal{Z}^S$, which could essentially be a one-dimensional manifold. This will clearly not be the case if $\boldsymbol{f}$ is an arbitrary function, necessitating constraints on its function class to make any generalization guarantees.

## 3  COMPOSITIONAL GENERALIZATION IN THEORY

In this section, we show theoretically how the ground-truth generator $\boldsymbol{f}$ and autoencoder $(\hat{\boldsymbol{g}}, \hat{\boldsymbol{f}})$ can be constrained to address both *slot identifiability* and *generalization*, thereby facilitating compositional generalization (complete proofs and further details are provided in App. A).

To address the problem of slot identifiability, Brady et al. (2023) proposed to constrain the generator $\boldsymbol{f}$ via assumptions on its Jacobian, which they called *compositionality* and *irreducibility*. Informally, compositionality states that each image pixel is *locally* a function of at most one latent slot, while irreducibility states that pixels belonging to the same object share information. We relegate a formal definition of irreducibility to App. A.4 and only restate the definition for compositionality.

**Definition 4** (Compositionality).  *A differentiable $\boldsymbol{f} : \mathcal{Z} \to \mathbb{R}^N$ is called compositional in $\boldsymbol{z} \in \mathcal{Z}$ if*

$$\frac{\partial \boldsymbol{f}_n}{\partial \boldsymbol{z}_k}(\boldsymbol{z}) \neq 0 \implies \frac{\partial \boldsymbol{f}_n}{\partial \boldsymbol{z}_j}(\boldsymbol{z}) = 0, \quad \text{for any } k, j \in [K], k \neq j \text{ and any } n \in [N]. \tag{4}$$

For a generator $\boldsymbol{f}$ satisfying these assumptions on $\mathcal{Z}$, Brady et al. (2023) showed that an autoencoder $(\hat{\boldsymbol{g}}, \hat{\boldsymbol{f}})$ with a compositional decoder (Def. 4) will slot-identify $\boldsymbol{z}$ on $\mathcal{Z}$ w.r.t. $\boldsymbol{f}$. This result is appealing for addressing Challenge 1 since it does not rely on assumptions on $p_{\boldsymbol{z}}$; however, it requires that the training space $\mathcal{X}^S$ is generated from the entire latent space $\mathcal{Z}$. We here show an extension for cases when the training space $\mathcal{X}^S$ arises from a convex, slot-supported *subset* $\mathcal{Z}^S$.

**Theorem 1** (Slot identifiability on slot-supported subset).  *Let $\boldsymbol{f} : \mathcal{Z} \to \mathcal{X}$ be a compositional and irreducible diffeomorphism. Let $\mathcal{Z}^S$ be a convex, slot-supported subset of $\mathcal{Z}$. An autoencoder $(\hat{\boldsymbol{g}}, \hat{\boldsymbol{f}})$*

*that minimizes $\mathcal{L}_{rec}(\mathcal{X}^S)$ for $\mathcal{X}^S = \boldsymbol{f}(\mathcal{Z}^S)$ and whose decoder $\hat{\boldsymbol{f}}$ is compositional on $\hat{\boldsymbol{g}}(\mathcal{X}^S)$, slot-identifies $\boldsymbol{z}$ on $\mathcal{Z}^S$ w.r.t. $\boldsymbol{f}$ in the sense of Def. 2.*

Thm. 1 solves Challenge 1 of slot identifiability on the slot-supported subset $\mathcal{Z}^S$, but to generalize compositionally, we still need to address Challenge 2 and extend this to all of $\mathcal{Z}$. Because we have slot identifiability on $\mathcal{Z}^S$, we know each ground-truth slot and corresponding inferred slot are related by a diffeomorphism $\boldsymbol{h}_k$. Since $\boldsymbol{h}_k$ is defined for all configurations of slot $\boldsymbol{z}_{\pi(k)}$, the representation which slot-identifies $\boldsymbol{z} = (\boldsymbol{z}_1, \ldots, \boldsymbol{z}_K)$ for any combination of slots (ID or OOD) in $\mathcal{Z}$ is given by

$$\boldsymbol{z}' = \big(\boldsymbol{h}_1(\boldsymbol{z}_{\pi(1)}), \ldots, \boldsymbol{h}_K(\boldsymbol{z}_{\pi(K)})\big), \quad \mathcal{Z}' = \boldsymbol{h}_1(\mathcal{Z}_{\pi(1)}) \times \cdots \times \boldsymbol{h}_K(\mathcal{Z}_{\pi(K)}). \quad (5)$$

Therefore, for an autoencoder to generalize its slot identifiability from $\mathcal{Z}^S$ to $\mathcal{Z}$, it should match this representation such that for any $\boldsymbol{z} \in \mathcal{Z}$,

$$\hat{\boldsymbol{g}}\big(\boldsymbol{f}(\boldsymbol{z})\big) = \boldsymbol{z}' \quad \text{and} \quad \hat{\boldsymbol{f}}(\boldsymbol{z}') = \boldsymbol{f}(\boldsymbol{z}). \quad (6)$$

We aim to satisfy these conditions by formulating properties of the decoder $\hat{\boldsymbol{f}}$ such that it fulfills the second condition, which we then leverage to regularize the encoder $\hat{\boldsymbol{g}}$ to fulfill the first condition.

## 3.1 DECODER GENERALIZATION VIA ADDITIVITY

We know from Thm. 1 that $\hat{\boldsymbol{f}}$ renders each inferred slot $\boldsymbol{h}_k(\boldsymbol{z}_{\pi(k)})$ correctly to a corresponding object in $\boldsymbol{x}$ for all possible values of $\boldsymbol{z}_{\pi(k)}$. Furthermore, because the generator $\boldsymbol{f}$ satisfies compositionality (Def. 4), we know that these *slot-wise renders* should not be affected by changes to the value of any other slot $\boldsymbol{z}_j$. This implies that for $\hat{\boldsymbol{f}}$ to satisfy Eq. 6, we only need to ensure that its slot-wise renders remain invariant when constructing $\boldsymbol{z}'$ with an OOD combination of slots $\boldsymbol{z}$. We show below that *additive* decoders can achieve this invariance.

**Definition 5** (Additive decoder). For an autoencoder $(\hat{\boldsymbol{g}}, \hat{\boldsymbol{f}})$ the decoder $\hat{\boldsymbol{f}}$ is said to be *additive* if

$$\hat{\boldsymbol{f}}(\boldsymbol{z}) = \sum_{k=1}^K \boldsymbol{\varphi}_k(\hat{\boldsymbol{z}}_k), \quad \text{where } \boldsymbol{\varphi}_k : \mathbb{R}^M \to \mathbb{R}^N \text{ for any } k \in [K] \text{ and } \hat{\boldsymbol{z}} \in \mathbb{R}^{KM}. \quad (7)$$

We can think of an additive decoder $\hat{\boldsymbol{f}}$ as rendering each slot $\hat{\boldsymbol{z}}_k$ to an intermediate image via *slot functions* $\boldsymbol{\varphi}_k$, then summing these images to create the final output. These decoders are expressive enough to represent compositional generators (see App. A.7). Intuitively, they globally remove interactions between slots such that the correct renders learned on inferred latents of $\mathcal{Z}^S$ are propagated to inferred latents of the entire $\mathcal{Z}$. This is formalized with the following result.

**Theorem 2** (Decoder generalization). *Let $\boldsymbol{f} : \mathcal{Z} \to \mathcal{X}$ be a compositional diffeomorphism and $\mathcal{Z}^S$ be a slot-supported subset of $\mathcal{Z}$. Let $(\hat{\boldsymbol{g}}, \hat{\boldsymbol{f}})$ be an autoencoder that slot-identifies $\boldsymbol{z}$ on $\mathcal{Z}^S$ w.r.t. $\boldsymbol{f}$. If the decoder $\hat{\boldsymbol{f}}$ is additive, then it generalizes in the following sense: $\hat{\boldsymbol{f}}(\boldsymbol{z}') = \boldsymbol{f}(\boldsymbol{z})$ for any $\boldsymbol{z} \in \mathcal{Z}$, where $\boldsymbol{z}'$ is defined according to Eq. 5.*

Consequently, $\hat{\boldsymbol{f}}$ is now injective on $\mathcal{Z}'$ and we get $\hat{\boldsymbol{f}}(\mathcal{Z}') = \boldsymbol{f}(\mathcal{Z}) = \mathcal{X}$.

## 3.2 ENCODER GENERALIZATION VIA COMPOSITIONAL CONSISTENCY

Because the decoder $\hat{\boldsymbol{f}}$ generalizes such that $\hat{\boldsymbol{f}}(\boldsymbol{z}') = \boldsymbol{f}(\boldsymbol{z})$ (Thm. 2) and $\hat{\boldsymbol{f}}(\mathcal{Z}') = \mathcal{X}$, the condition on the encoder from Eq. 6 corresponds to enforcing that $\hat{\boldsymbol{g}}$ inverts $\hat{\boldsymbol{f}}$ on all of $\mathcal{X}$. This is ensured ID on the training space $\mathcal{X}^S$ by minimizing the reconstruction objective $\mathcal{L}_{rec}$ (Eq. 3). However, there is nothing enforcing that $\hat{\boldsymbol{g}}$ also inverts $\hat{\boldsymbol{f}}$ OOD outside of $\mathcal{X}^S$ (see Fig. 1B for a visualization of this problem). To address this, we propose the following regularizer.

**Definition 6** (Compositional consistency). Let $q_{\boldsymbol{z}'}$ be a distribution with $\mathrm{supp}(q_{\boldsymbol{z}'}) = \mathcal{Z}'$ (Eq. 5). An autoencoder $(\hat{\boldsymbol{g}}, \hat{\boldsymbol{f}})$ is said to be *compositionally consistent* if it minimizes the *compositional consistency loss*

$$\mathcal{L}_{\mathrm{cons}}(\hat{\boldsymbol{g}}, \hat{\boldsymbol{f}}, \mathcal{Z}') = \mathbb{E}_{\boldsymbol{z}' \sim q_{\boldsymbol{z}'}} \left[ \big\| \hat{\boldsymbol{g}}\big(\hat{\boldsymbol{f}}(\boldsymbol{z}')\big) - \boldsymbol{z}' \big\|_2^2 \right]. \quad (8)$$

The loss can be understood as first sampling an OOD combination of slots $z'$ by composing inferred ID slots $h_k(z_{\pi(k)})$. The decoder can then render $z'$ to create an OOD sample $\hat{f}(z')$. Re-encoding this sample such that $\hat{g}(\hat{f}(z')) = z'$ then regularizes the encoder to invert the decoder OOD. We discuss how this regularization can be implemented in practice in Sec. 4.

### 3.3 Putting it All Together

Thm. 1 showed how slot identifiability can be achieved ID on $\mathcal{Z}^S$ if $\hat{f}$ satisfies compositionality, and Thm. 2, Def. 6 showed how this identifiability can be generalized to all of $\mathcal{Z}$ if the decoder is additive and compositional consistency is minimized. Putting these results together, we can now prove conditions for which an autoencoder will generalize compositionally (see Fig. 2 for an overview).

**Theorem 3** (Compositionally generalizing autoencoder). *Let $f : \mathcal{Z} \to \mathcal{X}$ be a compositional and irreducible diffeomorphism. Let $\mathcal{Z}^S$ be a convex, slot-supported subset of $\mathcal{Z}$. Let $(\hat{g}, \hat{f})$ be an autoencoder with additive decoder $\hat{f}$ (Def. 5). If $\hat{f}$ is compositional on $\hat{g}(\mathcal{X}^S)$ and $\hat{g}, \hat{f}$ solve*

$$\mathcal{L}_{rec}(\hat{g}, \hat{f}, \mathcal{X}^S) + \lambda \mathcal{L}_{cons}(\hat{g}, \hat{f}, \mathcal{Z}') = 0, \qquad \text{for some } \lambda > 0, \tag{9}$$

*then the autoencoder $(\hat{g}, \hat{f})$ generalizes compositionally w.r.t. $\mathcal{Z}^S$ in the sense of Def. 3.*

*Moreover, $\hat{g} : \mathcal{X} \to \hat{\mathcal{Z}}$ inverts $\hat{f} : \mathcal{Z}' \to \mathcal{X}$ and also $\hat{\mathcal{Z}} = \mathcal{Z}' = h_1(\mathcal{Z}_{\pi(1)}) \times \cdots \times h_K(\mathcal{Z}_{\pi(K)})$.*

## 4 Compositional Generalization in Practice

**Compositionality** Thm. 3 explicitly assumes that the decoder $\hat{f}$ satisfies compositionality on $\hat{g}(\mathcal{X}^S)$ but does not give a recipe to enforce this in practice. Brady et al. (2023) proposed a regularizer that enforces compositionality if minimized (see App. B.4), but their objective is computationally infeasible to optimize for larger models, thus limiting its practical use. At the same time, Brady et al. (2023) showed that explicitly optimizing this objective may not always be necessary, as the object-centric models used in their experiments seemed to minimize it implicitly, likely through the inductive biases in these models. We observe a similar phenomenon (see Fig. 4, right) and thus rely on these inductive biases to satisfy compositionality in our experiments in Sec. 6.

**Additivity** It is trivial to implement an additive decoder by parameterizing the slot functions $\varphi_k$ from Eq. 7 as, e.g., deconvolution neural networks. This resembles the decoders typically used in object-centric learning, with the key difference being the use of *slot-wise masks* $m_k$. Specifically, existing models commonly use a decoder of the form

$$\hat{f}(z) = \sum_{k=1}^{K} \tilde{m}_k \odot x_k, \qquad \tilde{m}_k = \sigma(m)_k, \qquad (m_k, x_k) = \varphi_k(z_k), \tag{10}$$

where $\odot$ is an element-wise multiplication and $\sigma(\cdot)$ denotes the softmax function. Using masks $m_k$ in this way facilitates modeling occluding objects but violates additivity as the softmax normalizes masks across slots, thus introducing interactions between slots during rendering. We empirically investigate how this interaction affects compositional generalization in Sec. 6.2.

**Compositional Consistency** The main challenge with implementing the proposed compositional consistency loss $\mathcal{L}_{\text{cons}}$ (Def. 6) is sampling $z'$ from $q_{z'}$ with support over $\hat{\mathcal{Z}}$. First, note that we defined $\mathcal{Z}'$ in Eq. 5 through use of the functions $h_k$, but can equivalently write

$$h_k(z_{\pi(k)}) = \hat{g}_k(f(z)) \quad \text{and} \quad \mathcal{Z}' = \hat{g}_1(\mathcal{X}^S) \times \cdots \times \hat{g}_K(\mathcal{X}^S). \tag{11}$$

The reformulation highlights that we can construct OOD samples in the consistency regularization from ID observations by randomly shuffling the slots of two inferred ID latents $\hat{z}^{(1)}, \hat{z}^{(2)}$ via $\rho_k \sim \mathcal{U}\{1, 2\}$. Because $\mathcal{Z}^S$ is a slot-supported subset, constructing $z'$ as

$$z' = (\hat{z}_1^{(\rho_1)}, \ldots, \hat{z}_K^{(\rho_K)}), \quad \text{where for } i \in \{1, 2\} \quad \hat{z}^{(i)} = \hat{g}(x^{(i)}), \ x^{(i)} \sim p_x \tag{12}$$

ensures that the samples $z'$ cover the entire $\mathcal{Z}'$. Practically, we sample $\hat{z}^{(1)}, \hat{z}^{(2)}$ uniformly from the current batch. The compositional consistency objective with this sampling strategy is illustrated

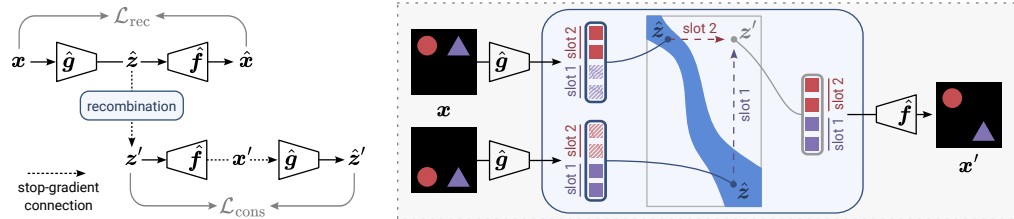

Figure 3: **Compositional consistency regularization**. In addition to the reconstruction objective, $\mathcal{L}_{\text{cons}}$ is minimized on recombined latents $z'$. Recombining slots of the inferred latents $\hat{z}$ of two ID samples produces a latent $z'$, which can be rendered to an OOD sample $x'$ due to the decoder $\hat{f}$ generalizing OOD. The encoder $\hat{g}$ is optimized to re-encode this sample to match $z'$.

in Fig. 3. Note that the order of slots is generally not preserved between $\hat{g}\big(\hat{f}(z')\big)$ and $z'$ so that we pair slots using the Hungarian algorithm (Kuhn, 1955) before calculating the loss. Furthermore, enforcing the consistency loss can be challenging in practice if the encoder contains stochastic operations such as the random re-initialization of slots in the Slot Attention module (Locatello et al., 2020a) during each forward pass. We explore the impact of this in Sec. 6.2.

## 5 RELATED WORK

**Theoretical Analyses of Compositional Generalization**    Prior works have addressed identifiability and generalization theoretically in isolation. For example, several results show how identifiability can be achieved through assumptions on the latent distribution Hyvärinen and Morioka (2016; 2017); Hyvärinen et al. (2019); Khemakhem et al. (2020a;b); Shu et al. (2020); Locatello et al. (2020b); Gresele et al. (2019); Lachapelle et al. (2021); Klindt et al. (2021); Hälvä et al. (2021); von Kügelgen et al. (2021); Liang et al. (2023) or via structural assumptions on the generator function (Gresele et al., 2021; Horan et al., 2021; Moran et al., 2022; Buchholz et al., 2022; Zheng et al., 2022; Brady et al., 2023). However, none of these deal with generalization. On the other hand, frameworks for OOD generalization were proposed in the context of object-centric world models (Zhao et al., 2022) and regression problems (e.g., Netanyahu et al., 2023), with latent variable formulations that closely resemble our work. In this context, OOD generalization was proven for additive inference models (Dong and Ma, 2022) or slot-wise functions composed with a known nonlinearity (Wiedemer et al., 2023). Yet, these results are formulated in a regression setting, which assumes the problem of identifiability is solved a priori. Concurrent work from Lachapelle et al. (2023) also considers identifiability and generalization. Similar to us, they leverage additivity to achieve decoder generalization and show that additivity is sufficient for identifiability under additional assumptions on the decoder, while allowing more general supports. However, they only focus on decoder generalization, while we show theoretically and empirically how to enforce that the encoder also generalizes OOD.

**Compositional Consistency Regularization**    Our compositional consistency loss (Def. 6), which generates and trains on novel data by composing previously learned concepts, resembles ideas in both natural and artificial intelligence. In natural intelligence, (Schwartenbeck et al., 2021; Kurth-Nelson et al., 2022; Bakermans et al., 2023) propose that the hippocampal formation implements a form of compositional replay in which new knowledge is derived by composing previously learned abstractions. In machine learning, prior works Rezende and Viola (2018); Cemgil et al. (2020); Sinha and Dieng (2021); Leeb et al. (2022) have shown that an encoder can fail to correctly encode samples generated by a decoder, though not in the context of compositional generalization. For program synthesis, Ellis et al. (2023) propose training a recognition model on compositions of learned programs. In object-centric learning, Assouel et al. (2022) also train an encoder using on images from recomposed slots; however, the model is tailored to a specific visual reasoning task.

## 6 EXPERIMENTS

This section first verifies our main theoretical result (Thm. 3) on synthetic multi-object data (Sec. 6.1). We then ablate the impact of each of our theoretical assumptions on compositional generalization using the object-centric model Slot Attention (Locatello et al., 2020a) (Sec. 6.2).

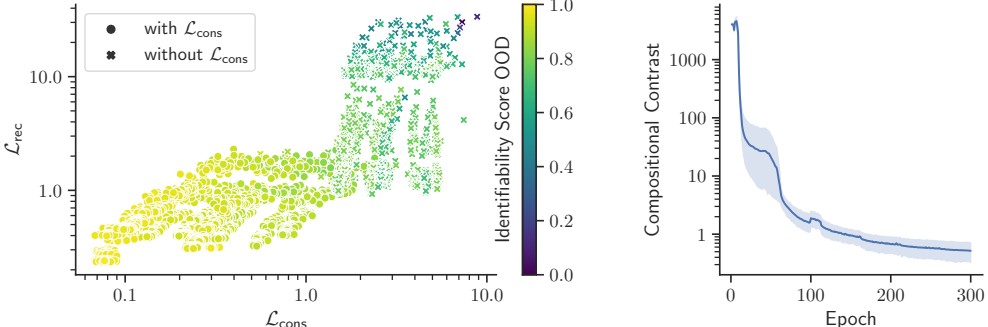

Figure 4: **Experimental validation of Thm. 3**. **Left**: Slot identifiability is measured throughout training as a function of reconstruction loss ($\mathcal{L}_{rec}$, Eq. 3) and compositional consistency ($\mathcal{L}_{cons}$, Def. 6). As predicted by Thm. 3, models which minimize $\mathcal{L}_{rec}$ and $\mathcal{L}_{cons}$ learn representations that are slot identifiable OOD. **Right**: Compositional contrast (see App. B.4) decreases throughout training, indicating that the decoder is implicitly optimized to be compositional (Def. 4).

**Data** We generate multi-object data using the Spriteworld renderer (Watters et al., 2019). Images contain two objects on a black background (Fig. 6), each specified by four continuous latents (x/y position, size, color) and one discrete latent (shape). To ensure that the generator satisfies compositionality (Def. 4), we exclude images with occluding objects. Irreducibility is almost certainly satisfied due to the high dimensionality of each image, as argued in (Brady et al., 2023). We sample latents on a slot-supported subset by restricting support to a diagonal strip in $\mathcal{Z}$ (see App. B.1).

**Metrics** To measure a decoder's compositionality (Def. 4), we rely on the compositional contrast regularizer from Brady et al. (2023) (App. B.4), which was proven to be zero if a function is compositional. To measure slot identifiability, we follow Locatello et al. (2020a); Dittadi et al. (2021); Brady et al. (2023) and fit nonlinear regressors to predict each ground-truth slot $z_i$ from an inferred slot $\hat{z}_j$ for every possible pair of slots. The regressor's fit measured by $R^2$ score quantifies how much information $\hat{z}_j$ encodes about $\hat{z}_i$. We subsequently determine the best assignment between slots using the Hungarian algorithm (Kuhn, 1955) and report the $R^2$ averaged over all matched slots.

## 6.1 VERIFYING THE THEORY

**Experimental Setup** We train an autoencoder with an additive decoder on the aforementioned multi-object dataset. The model uses two latent slots with 6 dimensions each. We train the model to minimize the reconstruction loss $\mathcal{L}_{rec}$ (Eq. 3) for 100 epochs, then introduce the compositional consistency loss $\mathcal{L}_{cons}$ (Def. 6) and jointly optimize both objectives for an additional 200 epochs.

**Results** Fig. 4 (Right) shows that compositional contrast decreases over the course of training without additional regularization, thus justifying our choice not to optimize it explicitly. Fig. 4 (Left) visualizes slot identifiability of OOD latents as a function of $\mathcal{L}_{rec}$ and $\mathcal{L}_{cons}$. OOD slot identifiability is maximized exactly when $\mathcal{L}_{rec}$ and $\mathcal{L}_{cons}$ are minimized, as predicted by Thm. 3. This is corroborated by the heatmaps in Fig. 1A-C, which illustrate that additivity enables the decoder to generalize as predicted by Thm. 2 but minimizing $\mathcal{L}_{cons}$ is required for the encoder to also generalize OOD.

## 6.2 ABLATING IMPACT OF THEORETICAL ASSUMPTIONS

**Experimental Setup** While Sec. 6.1 showed that our theoretical assumption can empirically enable compositional generalization, these assumptions differ from typical practices in object-centric models. We, therefore, ablate the relevance of each assumption for compositional generalization in the object-centric model Slot Attention. We investigate the effect of additivity by comparing a non-additive decoder that normalizes masks across slots using a softmax with an additive decoder that replaces the softmax with slot-wise sigmoid functions. We also train the model with and without optimizing compositional consistency. Finally, we explore the impact of using a deterministic encoder by replacing Slot Attention's random initialization of slots with a fixed initialization. All models use two slots with 16 dimensions each and are trained on the multi-object dataset from Sec. 6.1.

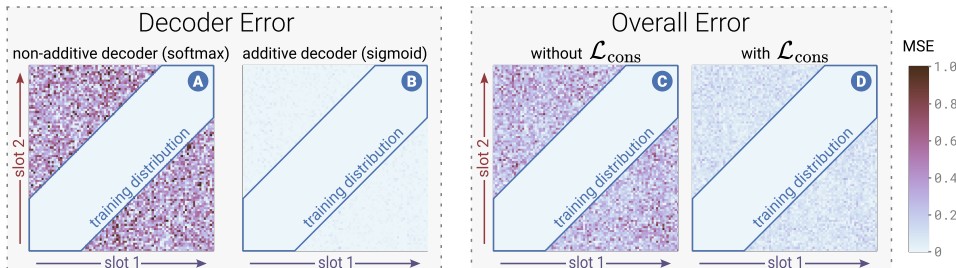

Figure 5: **Compositional generalization for Slot Attention**. Visualizing the decoder reconstruction error over a 2D projection of the latent space (see App. B.3 for details) reveals that the non-additive masked decoder in Slot Attention does not generalize OOD on our dataset (**A**). Making the decoder additive by replacing softmax mask normalization with slot-wise sigmoid functions makes the decoder additive and enables OOD generalization (**B**, Thm. 2). The full model does not generalize compositionally, however, since the encoder fails to invert the decoder OOD (**C**). Regularizing with the compositional consistency loss addresses this, enabling generalization (**D**, Thm. 3).

Table 1: **Compositional generalization for Slot Attention** in terms of slot identifiability and reconstruction quality. Both metrics are close to optimal ID but degrade OOD with the standard assumptions in Slot Attention. Incorporating decoder additivity (Add.), compositional consistency ($\mathcal{L}_{cons}$), and deterministic inference (Det.) improves OOD performance.

| Add. | $\mathcal{L}_{cons}$ | Det. | Identifiability $R^2\uparrow$ | | Reconstruction $R^2\uparrow$ | |
| | | | ID | OOD | ID | OOD |
|---|---|---|---|---|---|---|
| ✗ | ✗ | ✗ | $0.99_{\pm1.7e-3}$ | $0.81_{\pm9.0e-2}$ | $0.99_{\pm1.0e-4}$ | $0.71_{\pm1.9e-2}$ |
| ✓ | ✗ | ✗ | $0.99_{\pm2.3e-3}$ | $0.83_{\pm5.4e-2}$ | $0.99_{\pm5.8e-4}$ | $0.72_{\pm2.1e-2}$ |
| ✓ | ✓ | ✗ | $0.99_{\pm2.9e-2}$ | $0.92_{\pm3.4e-2}$ | $0.99_{\pm8.3e-4}$ | $0.79_{\pm7.2e-2}$ |
| ✓ | ✓ | ✓ | $0.99_{\pm1.9e-3}$ | $\mathbf{0.94}_{\pm2.2e-2}$ | $0.99_{\pm1.9e-4}$ | $\mathbf{0.92}_{\pm2.1e-2}$ |

**Results** Fig. 5 illustrates that the non-additive decoder in Slot Attention does not generalize OOD on our multi-object dataset. Moreover, regularization via the consistency loss is required to make the encoder generalize. Tab. 1 ablates the effect of these assumptions for Slot Attention. We see that satisfying additivity and compositional consistency and making inference deterministic (see Sec. 4) improves OOD identifiability and reconstruction performance.

## 7 DISCUSSION

**Extensions of Experiments** Our experiments in Sec. 6.2 provide evidence that compositional generalization will not emerge naturally in object-centric models such as Slot Attention. However, to gain a more principled understanding of the limits of compositional generalization in these models, experiments with a broader set of architectures on more datasets are required. Additionally, $\mathcal{L}_{cons}$, as implemented in Sec. 6, samples novel slot combinations in a naive uniform manner, potentially giving rise to implausible images in more complex settings, e.g., by sampling two background slots. Thus, a more principled sampling scheme should be employed to scale this loss.

**Extensions of Theory** The assumptions of *compositionality* and *additivity* on the decoder make progress towards a theoretical understanding of compositional generalization, yet are inherently limiting. Namely, they do not allow slots to interact during rendering and thus cannot adequately model general multi-object scenes or latent abstractions outside of objects. Thus, a key direction is to understand how slot interactions can be introduced while maintaining compositional generalization.

**Conclusion** Compositional generalization is crucial for robust machine perception; however, a principled understanding of how it can be realized has been lacking. We show in object-centric learning that compositional generalization is theoretically and empirically possible for autoencoders that possess a structured decoder and regularize the encoder to invert the decoder OOD. While our results do not provide an immediate recipe for compositional generalization in real-world object-centric learning, they lay the foundation for future theoretical and empirical works.

## Reproducibility Statement

Detailed versions of all theorems and definitions, as well as the full proofs for all results are included in App. A. We attach our codebase to facilitate the reproduction of our experiments. All hyperparameters, model architectures, training regimes, datasets, and evaluation metrics are provided in the codebase. Explanations for design choices are given in Sec. 6 in the main text and App. B. The implementation of the compositional consistency loss is detailed in Sec. 4, paragraph 3.

## Contributions

TW, JB, and WB initiated the project. JB and TW led the project. AP conducted all experiments with advising from TW and JB. AJ, JB, and TW conceived the conceptual ideas behind the theorems. AJ developed the theoretical results and presented them in App. A. TW and JB led the writing of the manuscript with contributions to the theory presentation from AJ and additional contributions from AP, WB, and MB. TW created Figs. 1, 2, and 3 with insights from all authors. AP and TW created Figs. 4 and 5 with insights from JB.

## Acknowledgements

The authors would like to thank (in alphabetical order): Andrea Dittadi, Egor Krasheninnikov, Evgenii Kortukov, Julius von Kügelgen, Prasanna Mayilvahannan, Roland Zimmermann, Sébastien Lachapelle, and Thomas Kipf for helpful insights and discussions.

This work was supported by the German Federal Ministry of Education and Research (BMBF): Tübingen AI Center, FKZ: 01IS18039A, 01IS18039B. WB acknowledges financial support via an Emmy Noether Grant funded by the German Research Foundation (DFG) under grant no. BR 6382/1-1 and via the Open Philantropy Foundation funded by the Good Ventures Foundation. WB is a member of the Machine Learning Cluster of Excellence, EXC number 2064/1 – Project number 390727645. This research utilized compute resources at the Tübingen Machine Learning Cloud, DFG FKZ INST 37/1057-1 FUGG. The authors thank the International Max Planck Research School for Intelligent Systems (IMPRS-IS) for supporting TW and AJ.

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

Table 2: **General notation and nomenclature**.

| | |
|---:|:---|
| $K$ | number of slots |
| $M$ | dimensionality of slots |
| $N$ | dimensionality of observations |
| $[n]$ | the set $\{1, 2, \ldots, n\}$ |
| $\boldsymbol{f} : A \rightarrowtail B$ | function $\boldsymbol{f}$ is defined on a subset of $A$, i.e. $\mathrm{dom}(\boldsymbol{f}) \subseteq A$ |
| $\boldsymbol{f} \in C^k(A, B)$ | function $\boldsymbol{f}$ defined on $A$ with values in $B$ is $k$-times continuously differentiable |
| $\boldsymbol{f} \in C^k\mathrm{Diffeo}(A, B)$ | function $\boldsymbol{f}$ is a $C^k$-diffeomorphism around $A$ with values in $B$ (Def. 7) |
| $\mathrm{D}\boldsymbol{f}(\boldsymbol{x})$ | (total) derivative of function $\boldsymbol{f}$ in point $\boldsymbol{x}$ |
| $\mathrm{Lin}(V, W)$ | space of all linear operators between linear spaces $V$ and $W$ |
| $\tilde{\mathcal{Z}}$ (and $\tilde{\mathcal{X}}$) | a generic subset of $\mathbb{R}^{KM}$ (and $\mathbb{R}^N$), whose role will usually be fulfilled by either $\mathcal{Z}^S$ or $\mathcal{Z}$ (and $\mathcal{X}^S$ or $\mathcal{X}$) |
| $\mathcal{Z}_k^S$ | projection of $\mathcal{Z}^S$ onto the $k$-th slot space $\mathcal{Z}_k$ |
| $\partial_k \boldsymbol{f}_n(\boldsymbol{z})$ | $\dfrac{\partial \boldsymbol{f}_n}{\partial \boldsymbol{z}_k}(\boldsymbol{z})$ |
| $I_k^{\boldsymbol{f}}(\boldsymbol{z})$ | $\left\{ n \in [N] \,\middle|\, \partial_k \boldsymbol{f}_n(\boldsymbol{z}) \neq 0 \right\}$ |
| $\boldsymbol{f}_S(\boldsymbol{z})$ | subvector of $\boldsymbol{f}(\boldsymbol{z})$ corresponding to coordinates $S \subseteq [N]$ |

## A  DEFINITIONS, THEOREMS, AND PROOFS

In this section, we detail the theoretical contributions of the paper, including all the proofs. Although there is no change in their contents, the formulation of some definitions and theorems are slightly altered here to be more precise and cover edge cases omitted in the main text. Hence, the numbering of the restated elements is reminiscent of that used in the main text.

Throughout the following subsections, let the *number of slots* $K$, the *slot dimensionality* $M$, and the *observed dimensionality* $N$ be arbitrary but fixed positive integers such that $N \geq KM$.

### A.1  INTRODUCTION AND DIFFEOMORPHISMS

This subsection aims to provide an accesible and completely formal definition of what we would call throughout Sec. 2 and 3 a *diffeomorphism*.

**Definition 7** ($C^k$-Diffeomorphism). Let $\tilde{\mathcal{Z}}$ be a closed subset of $\mathbb{R}^{KM}$. A function $\boldsymbol{f} : \mathbb{R}^{KM} \rightarrowtail \mathbb{R}^N$ is said to be a $C^k$-*diffeomorphism around* $\tilde{\mathcal{Z}}$, denoted by $\boldsymbol{f} \in C^k\mathrm{Diffeo}(\tilde{\mathcal{Z}}, \mathbb{R}^N)$, if

*i)* $\boldsymbol{f}$ is defined in an open neighbourhood $\mathcal{W} \subseteq \mathbb{R}^{KM}$ of $\tilde{\mathcal{Z}}$,

*ii)* $\boldsymbol{f}$ is $k$-times continuously differentiable on $\mathcal{W}$, i.e. $\boldsymbol{f} \in C^k(\mathcal{W})$,

*iii)* $\boldsymbol{f}$ is injective on $\tilde{\mathcal{Z}}$, i.e. bijective between $\tilde{\mathcal{Z}}$ and $\boldsymbol{f}(\tilde{\mathcal{Z}})$ and

*iv)* for any $\boldsymbol{z} \in \tilde{\mathcal{Z}}$ the derivative $\mathrm{D}\boldsymbol{f}(\boldsymbol{z}) \in \mathrm{Lin}(\mathbb{R}^{KM}, \mathbb{R}^N)$ is injective (or equivalently, of full column-rank).

**Remark.** In the special case when a function $\boldsymbol{h}$ is a diffeomorphism around $\tilde{\mathcal{Z}} \subseteq \mathbb{R}^{KM}$, but also maps to $\mathbb{R}^{KM}$, for any $\boldsymbol{z} \in \tilde{\mathcal{Z}}$ the derivative $\mathrm{D}\boldsymbol{h}(\boldsymbol{z})$ is a bijective, hence invertible linear transformation of $\mathbb{R}^{KM}$. Besides that, because of *iii)*, $\boldsymbol{h}$ is injective. Based on the inverse function theorem we may conclude that $\boldsymbol{h}$ is injective in an open neighbourhood $\mathcal{W}'$ of $\tilde{\mathcal{Z}}$ and $\boldsymbol{h}^{-1}|_{\boldsymbol{h}(\mathcal{W}')}$ is also $k$-times continuously differentiable (where, of course, $\boldsymbol{h}(\tilde{\mathcal{Z}}) \subseteq \boldsymbol{h}(\mathcal{W}')$).

Hence, we arrive to the more familiar definition of a *diffeomorphism*, i.e. $\boldsymbol{h}$ being bijective with both $\boldsymbol{h}$ and $\boldsymbol{h}^{-1}$ being continuously differentiable. However, the latter definition cannot be applied in the more general case when the dimensionality of the co-domain is larger than that of the domain, since $\boldsymbol{f}^{-1}|_{\boldsymbol{f}(\tilde{\mathcal{Z}})}$ cannot be differentiable on a lower dimensional submanifold $\boldsymbol{f}(\tilde{\mathcal{Z}})$.

With some variations, the mathematical object with the properties listed in Def. 7 is often called an *embedding* and serves as a generator or a parametrization of a lower dimensional submanifold of $\mathbb{R}^N$. This is closer to the interpretation we employ here.

**Remark.** The literature on diffeomorphisms often defines them as smooth, i.e. infinitely many times differentiable functions with smooth inverses. In our results and proofs, twice differentiability will suffice.

## A.2 Definition of generative process, autoencoder

Here we recall the definition of *slot-supported subsets* from Def. 1 and provide a definition of the latent variable model that encapsulates all of our assumptions on the generative process outlined in Sec. 2.

**Definition 8** (Projection onto a slot space)**.** For any $\mathcal{Z}^S \subseteq \mathcal{Z} = \mathcal{Z}_1 \times \cdots \times \mathcal{Z}_K$, let

$$\mathcal{Z}_k^S := \left\{ \boldsymbol{z}_k | \boldsymbol{z} \in \mathcal{Z}^S \right\} \tag{13}$$

be the projection of $\mathcal{Z}^S$ onto the $k$-th slot space $\mathcal{Z}_k$.

**Definition 1b** (Slot-supported subset)**.** Let $\mathcal{Z} = \mathcal{Z}_1 \times \cdots \times \mathcal{Z}_K$. A set $\mathcal{Z}^S \subseteq \mathcal{Z}$ is said to be a *slot-supported subset* of $\mathcal{Z}$ if

$$\mathcal{Z}_k^S \overset{Def.\,8}{=} \left\{ \boldsymbol{z}_k | \boldsymbol{z} \in \mathcal{Z}^S \right\} = \mathcal{Z}_k \text{ for any } k \in [K]. \tag{14}$$

**Definition 9** (Partially observed generative process, POGP)**.** A triplet $(\mathcal{Z}, \mathcal{Z}^S, \boldsymbol{f})$ is called a *partially observed generative process (POGP)*, if

   *i)* $\mathcal{Z} = \mathcal{Z}_1 \times \ldots \times \mathcal{Z}_K$ for convex, closed sets $\mathcal{Z}_k \subseteq \mathbb{R}^M$,

   *ii)* $\mathcal{Z}^S \subseteq \mathcal{Z}$ is a closed, slot-supported subset of $\mathcal{Z}$ (Def. 1b) and

   *iii)* $\boldsymbol{f} \in C^1 \mathrm{Diffeo}(\mathcal{Z}, \mathbb{R}^N)$ is a diffeomorphism around $\mathcal{Z}$ (in the sense of Def. 7).

For a given POGP $(\mathcal{Z}, \mathcal{Z}^S, \boldsymbol{f})$, we refer to $\mathcal{Z}$ as the *latent space* and $\mathcal{Z}^S$ as the *training latent space*. $\boldsymbol{f}$ is the *generator*, $\mathcal{X} := \boldsymbol{f}(\mathcal{Z})$ and $\mathcal{X}^S := \boldsymbol{f}(\mathcal{Z}^S)$ are the *data space* and *training space* respectively.

We also provide a definition for our object-centric model alongside the optimization objective.

**Definition 10** (Autoencoder, AE)**.** A pair $(\hat{\boldsymbol{g}}, \hat{\boldsymbol{f}})$ is called an *autoencoder (AE)*, if $\hat{\boldsymbol{g}} : \mathbb{R}^N \to \mathbb{R}^{KM}$ and $\hat{\boldsymbol{f}} : \mathbb{R}^{KM} \to \mathbb{R}^N$ are continuously differentiable functions, i.e. $\hat{\boldsymbol{g}}, \hat{\boldsymbol{f}} \in C^1$. We refer to the function $\hat{\boldsymbol{g}}$ as the *encoder* and $\hat{\boldsymbol{f}}$ as the *decoder*.

In case a POGP $(\mathcal{Z}, \mathcal{Z}^S, \boldsymbol{f})$ is given, we refer to $\hat{\mathcal{Z}} := \hat{\boldsymbol{g}}(\mathcal{X}) = \hat{\boldsymbol{g}}(\boldsymbol{f}(\mathcal{Z}))$ as the *inferred latent space* and $\hat{\mathcal{X}} := \hat{\boldsymbol{f}}(\hat{\mathcal{Z}})$ as the *reconstructed data space*.

If for a given set $\tilde{\mathcal{X}} \subseteq \mathbb{R}^N$ it holds that $\hat{\boldsymbol{g}} : \tilde{\mathcal{X}} \to \hat{\boldsymbol{g}}(\tilde{\mathcal{X}})$ is bijective and its inverse is $\hat{\boldsymbol{f}} : \hat{\boldsymbol{g}}(\tilde{\mathcal{X}}) \to \tilde{\mathcal{X}}$ (also invertible), then we say that the autoencoder $(\hat{\boldsymbol{g}}, \hat{\boldsymbol{f}})$ is *consistent* on $\tilde{\mathcal{X}}$.

**Definition 11** (Reconstruction Loss)**.** Let $(\hat{\boldsymbol{g}}, \hat{\boldsymbol{f}})$ be an autoencoder. Let $\tilde{\mathcal{X}} \subseteq \mathbb{R}^N$ be an arbitrary set and $p_{\boldsymbol{x}}$ be an arbitrary continuous distribution function with $\mathrm{supp}(p_{\boldsymbol{x}}) = \tilde{\mathcal{X}}$. The following quantity is called the *reconstruction loss of* $(\hat{\boldsymbol{g}}, \hat{\boldsymbol{f}})$ *with respect to* $p_{\boldsymbol{x}}$:

$$\mathcal{L}_{\mathrm{rec}}(\hat{\boldsymbol{g}}, \hat{\boldsymbol{f}}, p_{\boldsymbol{x}}) := \mathbb{E}_{\boldsymbol{x} \sim p_{\boldsymbol{x}}} \left[ \left\| \hat{\boldsymbol{f}}(\hat{\boldsymbol{g}}(\boldsymbol{x})) - \boldsymbol{x} \right\|_2^2 \right]. \tag{15}$$

The following simple lemma tells us that for $\mathcal{L}_{\mathrm{rec}}(\hat{\boldsymbol{g}}, \hat{\boldsymbol{f}}, p_{\boldsymbol{x}})$ to vanish, the exact choice of $p_{\boldsymbol{x}}$ is irrelevant, and that in this case the decoder (left) inverts the encoder on $\tilde{\mathcal{X}}$. In other words, the autoencoder is consistent on $\tilde{\mathcal{X}}$.

**Lemma 4** (Vanishing $\mathcal{L}_{\mathrm{rec}}$ implies invertibility on $\tilde{\mathcal{X}}$)**.** *Let* $(\hat{\boldsymbol{g}}, \hat{\boldsymbol{f}})$ *be an autoencoder and let* $\tilde{\mathcal{X}} \subseteq \mathbb{R}^N$ *be an arbitrary set. The following statements are equivalent:*

   *i) there exists a cont. distribution function $p_{\boldsymbol{x}}$ with $\mathrm{supp}(p_{\boldsymbol{x}}) = \tilde{\mathcal{X}}$ such that $\mathcal{L}_{rec}(\hat{\boldsymbol{g}}, \hat{\boldsymbol{f}}, p_{\boldsymbol{x}}) = 0$;*

   *ii) for any cont. distribution function $p_{\boldsymbol{x}}$ with $\mathrm{supp}(p_{\boldsymbol{x}}) = \tilde{\mathcal{X}}$ we have $\mathcal{L}_{rec}(\hat{\boldsymbol{g}}, \hat{\boldsymbol{f}}, p_{\boldsymbol{x}}) = 0$;*

   *iii) $\hat{\boldsymbol{f}} \circ \hat{\boldsymbol{g}}|_{\tilde{\mathcal{X}}} = id$ or, in other words, $(\hat{\boldsymbol{g}}, \hat{\boldsymbol{f}})$ is consistent on $\tilde{\mathcal{X}}$.*

**Notation** Since $\mathcal{L}_{\text{rec}}(\hat{\boldsymbol{g}}, \hat{\boldsymbol{f}}, p_{\boldsymbol{x}})$ vanishes either for all $p_{\boldsymbol{x}}$ or none at the same time, henceforth in the context of vanishing reconstruction loss, we are going to denote the *reconstruction loss of* $(\hat{\boldsymbol{g}}, \hat{\boldsymbol{f}})$ *w.r.t.* $p_{\boldsymbol{x}}$ as $\mathcal{L}_{\text{rec}}(\hat{\boldsymbol{g}}, \hat{\boldsymbol{f}}, \tilde{\mathcal{X}})$ or just $\mathcal{L}_{\text{rec}}(\tilde{\mathcal{X}})$.

*Proof of Lem. 4.* We prove the equivalence of these statements by proving $ii) \Rightarrow i) \Rightarrow iii) \Rightarrow ii)$.

The implication $ii) \Rightarrow i)$ is trivial.

For the implication $i) \Rightarrow iii)$, let us suppose that there exists a $p_{\boldsymbol{x}}$ with $\text{supp}(p_{\boldsymbol{x}}) = \tilde{\mathcal{X}}$ such that $\mathcal{L}_{\text{rec}}(\hat{\boldsymbol{g}}, \hat{\boldsymbol{f}}, p_{\boldsymbol{x}}) = 0$. Equivalently:

$$\mathbb{E}_{\boldsymbol{x} \sim p_{\boldsymbol{x}}} \left[ \left\| \hat{\boldsymbol{f}}(\hat{\boldsymbol{g}}(\boldsymbol{x})) - \boldsymbol{x} \right\|_2^2 \right] = 0. \tag{16}$$

This shows that $\left\| \hat{\boldsymbol{f}}(\hat{\boldsymbol{g}}(\boldsymbol{x})) - \boldsymbol{x} \right\|_2^2 = 0$ or $\hat{\boldsymbol{f}}(\hat{\boldsymbol{g}}(\boldsymbol{x})) = \boldsymbol{x}$ holds almost surely w.r.t. $\boldsymbol{x} \sim p_{\boldsymbol{x}}$. Since both $\hat{\boldsymbol{g}}, \hat{\boldsymbol{f}}$ are continuous functions, this implies that $\hat{\boldsymbol{f}} \circ \hat{\boldsymbol{g}}|_{\tilde{\mathcal{X}}} = id$, which is exactly what was to be proven.

Now, for $iii) \Rightarrow ii)$, let us suppose that $\hat{\boldsymbol{f}} \circ \hat{\boldsymbol{g}}|_{\tilde{\mathcal{X}}} = id$ and let $p_{\boldsymbol{x}}$ be any continuous distribution function with $\text{supp}(p_{\boldsymbol{x}}) = \tilde{\mathcal{X}}$. Since, $\hat{\boldsymbol{f}}(\hat{\boldsymbol{g}}(\boldsymbol{x})) = \boldsymbol{x}$ holds for any non-random $\boldsymbol{x} \in \tilde{\mathcal{X}}$, then with probability 1, $\hat{\boldsymbol{f}}(\hat{\boldsymbol{g}}(\boldsymbol{x})) = \boldsymbol{x}$ also holds. From this, we conclude that:

$$\mathcal{L}_{\text{rec}}(\hat{\boldsymbol{g}}, \hat{\boldsymbol{f}}) = \mathbb{E}_{\boldsymbol{x} \sim p_{\boldsymbol{x}}} \left[ \left\| \hat{\boldsymbol{f}}(\hat{\boldsymbol{g}}(\boldsymbol{x})) - \boldsymbol{x} \right\|_2^2 \right] = 0. \tag{17}$$

$\square$

## A.3 Definition of Slot Identifiability and Compositional Generalization

We are now ready to recall the definition of *slot identifiability* (Def. 2, originally in Brady et al. (2023)) in its most abstract form, followed by the definition of *compositional generalization* (Def. 3).

**Definition 2b** (Slot identifiability). Let $\boldsymbol{f} \in C^1 \text{Diffeo}(\mathcal{Z}, \mathbb{R}^N)$, where $\mathcal{Z} = \mathcal{Z}_1 \times \ldots \times \mathcal{Z}_K$ for closed sets $\mathcal{Z}_k \subseteq \mathbb{R}^M$, and let $\tilde{\mathcal{Z}} \subseteq \mathcal{Z}$ be a closed, slot-supported subset of $\mathcal{Z}$ (e.g. $\mathcal{Z}$ or $\mathcal{Z}^S$ from a POGP).

An autoencoder $(\hat{\boldsymbol{g}}, \hat{\boldsymbol{f}})$ is said to *slot-identify $\boldsymbol{z}$ on $\tilde{\mathcal{Z}}$ w.r.t. $\boldsymbol{f}$* via $\hat{\boldsymbol{h}}(\boldsymbol{z}) := \hat{\boldsymbol{g}}(\boldsymbol{f}(\boldsymbol{z}))$ if it minimizes $\mathcal{L}_{\text{rec}}(\tilde{\mathcal{X}})$ for $\tilde{\mathcal{X}} = \boldsymbol{f}(\tilde{\mathcal{Z}})$ and there exists a permutation $\pi$ of $[K]$ and a set of diffeomorphisms $\boldsymbol{h}_k \in C^1 \text{Diffeo}\left(\mathcal{Z}_{\pi(k)}, \hat{\boldsymbol{h}}_k(\tilde{\mathcal{Z}})\right)$ such that $\hat{\boldsymbol{h}}_k(\boldsymbol{z}) = \boldsymbol{h}_k(\boldsymbol{z}_{\pi(k)})$ for any $k$ and $\boldsymbol{z}$, where $\hat{\boldsymbol{h}}_k(\tilde{\mathcal{Z}})$ is the projection of the set $\hat{\boldsymbol{h}}(\tilde{\mathcal{Z}})$ (c.f. Def. 8).

**Definition 3b** (Compositional generalization). Let $(\mathcal{Z}, \mathcal{Z}^S, \boldsymbol{f})$ be a POGP. An autoencoder $(\hat{\boldsymbol{g}}, \hat{\boldsymbol{f}})$ that slot-identifies $\boldsymbol{z}$ on $\mathcal{Z}^S$ w.r.t. $\boldsymbol{f}$ is said to *generalize compositionally w.r.t. $\mathcal{Z}^S$*, if it also slot-identifies $\boldsymbol{z}$ on $\mathcal{Z}$ w.r.t. $\boldsymbol{f}$.

## A.4 Compositionality and irreducibility assumptions

In this subsection we present the formal definitions of *compositionality* and *irreducibility*, already mentioned in Sec. 3 and originally introduced in Brady et al. (2023). These two properties represent sufficient conditions for POGPs (Def. 9) to be slot-identifiable on the training latent space (Def. 2b).

Let $\boldsymbol{f} \in C^1(\mathcal{Z}, \mathbb{R}^N)$, $\mathcal{Z} \subseteq \mathbb{R}^{KM}$, $k \in [K]$, $\boldsymbol{z} \in \mathcal{Z}$. For the sake of brevity let us denote $\partial_k \boldsymbol{f}_n(\boldsymbol{z}) = \partial \boldsymbol{f}_n / \partial \boldsymbol{z}_k(\boldsymbol{z})$. In this case, let us define

$$I_k^{\boldsymbol{f}}(\boldsymbol{z}) = \left\{ n \in [N] \,\middle|\, \partial_k \boldsymbol{f}_n(\boldsymbol{z}) \neq 0 \right\} \tag{18}$$

the set of coordinates locally influenced (i.e., in point $\boldsymbol{z}$) by slot $\boldsymbol{z}_k$ w.r.t. the generator $\boldsymbol{f}$.

**Definition 4b** (Compositionality). We say that a function $\boldsymbol{f} \in C^1(\mathcal{Z}, \mathbb{R}^N)$, $\mathcal{Z} \subseteq \mathbb{R}^{KM}$, is *compositional on $\tilde{\mathcal{Z}} \subseteq \mathcal{Z}$* if for any $\boldsymbol{z} \in \tilde{\mathcal{Z}}$ and $k \neq j, k, j \in [N]$:

$$I_k^{\boldsymbol{f}}(\boldsymbol{z}) \cap I_j^{\boldsymbol{f}}(\boldsymbol{z}) = \varnothing. \tag{19}$$

Let $\boldsymbol{f}_S(\boldsymbol{z})$ denote the subvector of $\boldsymbol{f}(\boldsymbol{z})$ corresponding to coordinates $S \subseteq [N]$ and let $\mathrm{D}\boldsymbol{f}_S(z) \in \mathrm{Lin}(\mathbb{R}^{KM}, \mathbb{R}^{|S|})$ be the corresponding derivative.

**Definition 12** (Irreducibility). We say that a function $\boldsymbol{f} \in C^1(\mathcal{Z}, \mathbb{R}^N), \mathcal{Z} \subseteq \mathbb{R}^{KM}$ is *irreducible on* $\tilde{\mathcal{Z}} \subseteq \mathcal{Z}$ if for any $\boldsymbol{z} \in \tilde{\mathcal{Z}}$ and $k \in [N]$ and any non-trivial partition $I_k^{\boldsymbol{f}}(\boldsymbol{z}) = S_1 \cup S_2$ (i.e., $S_1 \cap S_2 = \varnothing$ and $S_1, S_2 \neq \varnothing$) we have:

$$\mathrm{rank}\left(\mathrm{D}\boldsymbol{f}_{S_1}(\boldsymbol{z})\right) + \mathrm{rank}\left(\mathrm{D}\boldsymbol{f}_{S_2}(\boldsymbol{z})\right) > \mathrm{rank}\left(\mathrm{D}\boldsymbol{f}_{I_k^{\boldsymbol{f}}(\boldsymbol{z})}(\boldsymbol{z})\right). \tag{20}$$

**Remark.** Given linear operators $A \in \mathrm{Lin}(U, V), B \in \mathrm{Lin}(U, W)$, the following upper bound holds for the rank of $(A, B) \in \mathrm{Lin}(U, V \times W)$:

$$\mathrm{rank}(A) + \mathrm{rank}(B) \geq \mathrm{rank}\left((A, B)\right). \tag{21}$$

Therefore, it holds in general that:

$$\mathrm{rank}\left(\mathrm{D}\boldsymbol{f}_{S_1}(z)\right) + \mathrm{rank}\left(\mathrm{D}\boldsymbol{f}_{S_2}(z)\right) \geq \mathrm{rank}\left(\mathrm{D}\boldsymbol{f}_{I_i^{\boldsymbol{f}}(z)}(z)\right), \tag{22}$$

hence, irreducibility only prohibits equality.

### A.5 Identifiability on the training latent space

Here we recall Thm. 1, originally presented in a slightly less general form in Brady et al. (2023).

**Theorem 1b** (Slot identifiability on slot-supported subset). *Let $(\mathcal{Z}, \mathcal{Z}^S, \boldsymbol{f})$ be a POGP (Def. 9) such that*

  *i) $\mathcal{Z}^S$ is convex and*

  *ii) $\boldsymbol{f}$ is compositional and irreducible on $\mathcal{Z}^S$.*

*Let $\left(\hat{\boldsymbol{g}}, \hat{\boldsymbol{f}}\right)$ be an autoencoder (Def. 10) such that*

  *iii) $\left(\hat{\boldsymbol{g}}, \hat{\boldsymbol{f}}\right)$ minimizes $\mathcal{L}_{rec}(\mathcal{X}^S)$ for $\mathcal{X}^S = \boldsymbol{f}(\mathcal{Z}^S)$ and*

  *iv) $\hat{\boldsymbol{f}}$ is compositional on $\hat{\mathcal{Z}}^S := \hat{\boldsymbol{g}}(\mathcal{X}^S)$.*

*Then $\left(\hat{\boldsymbol{g}}, \hat{\boldsymbol{f}}\right)$ slot-identifies $\boldsymbol{z}$ on $\mathcal{Z}^S$ w.r.t. $\boldsymbol{f}$ in the sense of Def. 2b.*

**Remark.** Due to Lem. 4, assumption *iii)* is equivalent to $\left(\hat{\boldsymbol{g}}, \hat{\boldsymbol{f}}\right)$ being consistent on $\mathcal{X}^S$, i.e., $\hat{\boldsymbol{g}}|_{\mathcal{X}^S}$ is injective and its inverse is $\hat{\boldsymbol{f}}|_{\hat{\mathcal{Z}}^S}$.

In its original framework, Brady et al. (2023) assumed the training latent space to be the entirety of $\mathbb{R}^{KM}$. In our case, however, $\mathcal{Z}^S$ is a closed, convex, slot-supported subset of $\mathcal{Z}$. Therefore, it is required to reprove Thm. 1b.

### A.6 Proof of slot identifiability on slot-supported subset

In this subsection we reprove Thm. 1b via 3 steps. First, we prove that the latent reconstruction function $\hat{\boldsymbol{h}} = \hat{\boldsymbol{g}} \circ \boldsymbol{f}$ is, under the consistency of the autoencoder, a diffeomorphism. Secondly, we restate Prop. 3 of Brady et al. (2023) using our notation. It is a result that locally describes the behaviour of $\hat{\boldsymbol{h}}$, irrespective of the shape of the latent training space. Hence, no proof is required. The third and final step is concluding Thm. 1b itself.

**Lemma 5** (Latent reconstruction is diffeomorphism). *Let $\boldsymbol{f} \in C^1 \mathrm{Diffeo}(\tilde{\mathcal{Z}}, \mathbb{R}^N)$ for $\tilde{\mathcal{Z}} \subseteq \mathbb{R}^{KM}$ closed and $\left(\hat{\boldsymbol{g}}, \hat{\boldsymbol{f}}\right)$ an autoencoder consistent on $\tilde{\mathcal{X}} := \boldsymbol{f}(\tilde{\mathcal{Z}})$. Then $\hat{\boldsymbol{h}} := \hat{\boldsymbol{g}} \circ \boldsymbol{f}$ is a $C^1$-diffeomorphism around $\tilde{\mathcal{Z}}$ (Def. 7).*

*In particular, for any $\boldsymbol{z} \in \tilde{\mathcal{Z}}$, $\mathrm{D}\hat{\boldsymbol{h}}(\boldsymbol{z})$ is invertible linear transformation of $\mathbb{R}^{KM}$, continuously depending on $\boldsymbol{z}$.*

*Proof.* Since $\boldsymbol{f}$ is $C^1$ in an open neighbourhood of $\tilde{\mathcal{Z}}$ and $\hat{\boldsymbol{g}}$ is $C^1$ on $\mathbb{R}^N$, it follows that $\hat{\boldsymbol{h}} = \hat{\boldsymbol{g}} \circ \boldsymbol{f}$ is also $C^1$ in an open neighbourhood of $\tilde{\mathcal{Z}}$.

The autoencoder $(\hat{\boldsymbol{g}}, \hat{\boldsymbol{f}})$ is consistent on $\tilde{\mathcal{X}}$, hence $\hat{\boldsymbol{g}}|_{\tilde{\mathcal{X}}}$ is injective and $\hat{\boldsymbol{h}} : \tilde{\mathcal{Z}} \to \hat{\boldsymbol{g}}(\tilde{\mathcal{X}})$ is bijective.

Moreover, the inverse of $\hat{\boldsymbol{g}}|_{\tilde{\mathcal{X}}}$ is $\hat{\boldsymbol{f}}$ restricted to $\hat{\boldsymbol{g}}(\tilde{\mathcal{X}})$. Therefore, $\hat{\boldsymbol{h}} = \hat{\boldsymbol{g}} \circ \boldsymbol{f}$ implies that for any $\boldsymbol{z} \in \tilde{\mathcal{Z}}$ it holds that $\hat{\boldsymbol{f}}(\hat{\boldsymbol{h}}(\boldsymbol{z})) = \boldsymbol{f}(\boldsymbol{z})$. After differentiation we receive that for any $\boldsymbol{z} \in \tilde{\mathcal{Z}}$:

$$\mathrm{D}\hat{\boldsymbol{f}}(\hat{\boldsymbol{h}}(\boldsymbol{z})) \, \mathrm{D}\hat{\boldsymbol{h}}(\boldsymbol{z}) = \mathrm{D}\boldsymbol{f}(\boldsymbol{z}). \tag{23}$$

However, $\boldsymbol{f}$ is a $C^1$-diffeomorphism, consequently $\mathrm{D}\boldsymbol{f}(\boldsymbol{z}) \in \mathrm{Lin}(\mathbb{R}^{KM}, \mathbb{R}^N)$ is injective. On the left-hand side, we then have an injective composition of linear functions. Therefore, $\mathrm{D}\hat{\boldsymbol{h}}(\boldsymbol{z}) \in \mathrm{Lin}(\mathbb{R}^{KM}, \mathbb{R}^{KM})$ is injective and, of course, bijective.

Consequently, $\hat{\boldsymbol{h}}$ is a $C^1$-diffeomorphism. $\qquad \square$

**Lemma 6** (Prop. 3 of Brady et al. (2023))**.** *Let $(\mathcal{Z}, \mathcal{Z}^S, \boldsymbol{f})$ POGP and $(\hat{\boldsymbol{g}}, \hat{\boldsymbol{f}})$ autoencoder satisfy the assumptions of Thm. 1b and let $\hat{\boldsymbol{h}} := \hat{\boldsymbol{g}} \circ \boldsymbol{f}$ (now a $C^1$-diffeomorphism around $\mathcal{Z}^S$ because of remark after Thm. 1b and Lem. 5).*

*Then for any $\boldsymbol{z} \in \mathcal{Z}^S$ and any $j \in [K]$ there exists a unique $k \in [K]$ such that $\partial_j \hat{\boldsymbol{h}}_k(\boldsymbol{z}) \neq 0$. For this particular $k$ it holds that $\partial_j \hat{\boldsymbol{h}}_k(\boldsymbol{z}) \in \mathrm{Lin}(\mathbb{R}^M, \mathbb{R}^M)$ is invertible.*

**Remark.** Since $\hat{\boldsymbol{h}}$ is a diffeomorphism, $\mathrm{D}\hat{\boldsymbol{h}}(\boldsymbol{z})$ is invertible. Thus, the statement of Lem. 6 is equivalent to saying that for any $\boldsymbol{z} \in \mathcal{Z}^S$ and any $k \in [K]$ there exists a unique $j \in [K]$ such that $\partial_j \hat{\boldsymbol{h}}_k(\boldsymbol{z}) \neq 0$.

*Proof of Thm. 1b.* Let $\hat{\boldsymbol{h}} := \hat{\boldsymbol{g}} \circ \boldsymbol{f}$. Based on Lem. 6 and the latest remark, for any $\boldsymbol{z}$ and any $k$ there exists unique $j$ such that $\partial_j \hat{\boldsymbol{h}}_k(\boldsymbol{z}) \neq 0$, and for this $j$, $\partial_j \hat{\boldsymbol{h}}_k(\boldsymbol{z}) \in \mathrm{Lin}(\mathbb{R}^{KM}, \mathbb{R}^{KM})$ is invertible.

**Step 1.** Firstly, we claim that, in this case, the mapping $k \mapsto j$ is bijective and independent of $\boldsymbol{z}$. More precisely, we show that there exists a permutation $\pi$ of $[K]$ such that

$$\text{for any } \boldsymbol{z}, k \text{ and } j: \quad \partial_j \hat{\boldsymbol{h}}_k(\boldsymbol{z}) \neq 0 \iff j = \pi(k) \tag{24}$$

and, in the latter case, $\partial_{\pi(k)} \hat{\boldsymbol{h}}_k(\boldsymbol{z})$ is invertible.

To prove this, we conclude from the invertibility of $\mathrm{D}\hat{\boldsymbol{h}}(\boldsymbol{z})$ that for any $\boldsymbol{z}$ there exists such a $\pi$. Now suppose that there exist two distinct points $\boldsymbol{z}^{(1)}, \boldsymbol{z}^{(2)} \in \mathcal{Z}^S$ and indices $k$ and $j_1 \neq j_2$ from $[K]$ such that $\partial_{j_1} \hat{\boldsymbol{h}}_k(\boldsymbol{z}^{(1)}) \neq 0, \partial_{j_2} \hat{\boldsymbol{h}}_k(\boldsymbol{z}^{(2)}) \neq 0$. Then, since $\mathcal{Z}^S$ is path-connected (as being convex), it provides us with a continuous function $\boldsymbol{\phi} : [0, 1] \to \mathcal{Z}^S$ such that $\boldsymbol{\phi}(0) = \boldsymbol{z}^{(1)}$ and $\boldsymbol{\phi}(1) = \boldsymbol{z}^{(2)}$. Let

$$t^* = \sup\{t \in [0, 1] \,|\, \partial_{j_1} \hat{\boldsymbol{h}}_k(\boldsymbol{\phi}(t)) \neq 0\}. \tag{25}$$

On one hand, $\partial_{j_2} \hat{\boldsymbol{h}}_k(\boldsymbol{\phi}(1)) = \partial_{j_2} \hat{\boldsymbol{h}}_k(\boldsymbol{z}^{(2)}) \neq 0$, hence $\partial_{j_1} \hat{\boldsymbol{h}}_k(\boldsymbol{\phi}(1)) = 0$ and for any $t > t^*$: $\partial_{j_1} \hat{\boldsymbol{h}}_k(\boldsymbol{\phi}(t)) = 0$. Therefore, because $\partial_{j_1} \hat{\boldsymbol{h}}_k \circ \boldsymbol{\phi}$ is continuous, it follows that $\partial_{j_1} \hat{\boldsymbol{h}}_k(\boldsymbol{\phi}(t^*)) = 0$.

On the other hand, $\partial_{j_1} \hat{\boldsymbol{h}}_k(\boldsymbol{\phi}(0)) = \partial_{j_1} \hat{\boldsymbol{h}}_k(\boldsymbol{z}^{(1)}) \neq 0$ implies that $t^* \neq 0$ and there exists a convergent sequence $(t_n) \subseteq [0, t^*)$ with $\lim_{n \to \infty} t_n = t^*$ such that $\partial_{j_1} \hat{\boldsymbol{h}}_k(\boldsymbol{\phi}(t_n)) \neq 0$. Therefore, for any $j \neq j_1$, $\partial_j \hat{\boldsymbol{h}}_k(\boldsymbol{\phi}(t_n)) = 0$. From the continuity of $\partial_j \hat{\boldsymbol{h}}_k \circ \boldsymbol{\phi}$ we conclude that for any $j \neq j_1$, $\partial_j \hat{\boldsymbol{h}}_k(\boldsymbol{\phi}(t^*)) = 0$.

Subsequently, for any $j \in [K]$ (either $j \neq j_1$ or $j = j_1$) it holds that $\partial_j \hat{\boldsymbol{h}}_k(\boldsymbol{\phi}(t^*)) = 0$. Thus, we get $\mathrm{D}\hat{\boldsymbol{h}}_k(\boldsymbol{\phi}(t^*)) = 0$, which contradicts $\hat{\boldsymbol{h}}$ being a diffeomorphism. Hence, there exists $\pi$ such that for any $\boldsymbol{z}, k$: $\partial_j \hat{\boldsymbol{h}}_k(\boldsymbol{z}) \neq 0 \Leftrightarrow j = \pi(k)$.

**Step 2.** Secondly, we now prove that $\hat{\boldsymbol{h}}$ acts slot-wise with permutation $\pi$, i.e. for any $k$ there exists $\boldsymbol{h}_k$ $C^1$-diffeomorphism around $\mathcal{Z}_{\pi(k)}$ such that for any $\boldsymbol{z}$, $\hat{\boldsymbol{h}}_k(\boldsymbol{z}) = \boldsymbol{h}_k(\boldsymbol{z}_{\pi(k)})$. By Def. 2b this would imply that $(\hat{\boldsymbol{g}}, \hat{\boldsymbol{f}})$ slot-identifies $\boldsymbol{z}$ on $\mathcal{Z}^S$ w.r.t. $\boldsymbol{f}$.

To see this, let $\boldsymbol{z}^{(1)}, \boldsymbol{z}^{(2)} \in \mathcal{Z}^S$ with $\boldsymbol{z}_{\pi(k)}^{(1)} = \boldsymbol{z}_{\pi(k)}^{(2)}$. To be proven: $\hat{\boldsymbol{h}}_k(\boldsymbol{z}^{(1)}) = \hat{\boldsymbol{h}}_k(\boldsymbol{z}^{(2)})$. Since $\mathcal{Z}^S$ is convex, the path $t \in [0, 1] \mapsto \boldsymbol{z}^{(t)} = \boldsymbol{z}^{(1)} + t \cdot (\boldsymbol{z}^{(2)} - \boldsymbol{z}^{(1)})$ is inside $\mathcal{Z}^S$. Then

$$\hat{\boldsymbol{h}}_k(\boldsymbol{z}^{(2)}) - \hat{\boldsymbol{h}}_k(\boldsymbol{z}^{(1)}) = \hat{\boldsymbol{h}}_k(\boldsymbol{z}^{(t)})\big|_0^1 = \int_0^1 \frac{d}{dt}[\hat{\boldsymbol{h}}_k(\boldsymbol{z}^{(t)})]dt \tag{26}$$

$$= \int_0^1 \mathrm{D}\hat{\boldsymbol{h}}_k(\boldsymbol{z}^{(t)})(\boldsymbol{z}^{(2)} - \boldsymbol{z}^{(1)})dt = \int_0^1 \sum_{j=1}^K \partial_j \hat{\boldsymbol{h}}_k(\boldsymbol{z}^{(t)})(\boldsymbol{z}_j^{(2)} - \boldsymbol{z}_j^{(1)})dt. \tag{27}$$

First using the fact that $\partial_j \hat{\boldsymbol{h}}_k(\boldsymbol{z}^{(t)}) \neq 0 \Leftrightarrow j = \pi(k)$ and then substituting $\boldsymbol{z}_{\pi(k)}^{(1)} = \boldsymbol{z}_{\pi(k)}^{(2)}$, we receive:

$$\hat{\boldsymbol{h}}_k(\boldsymbol{z}^{(2)}) - \hat{\boldsymbol{h}}_k(\boldsymbol{z}^{(1)}) = \int_0^1 \partial_{\pi(k)} \hat{\boldsymbol{h}}_k(\boldsymbol{z}^{(t)})(\boldsymbol{z}_{\pi(i)}^{(2)} - \boldsymbol{z}_{\pi(i)}^{(1)})dt = 0. \tag{28}$$

$\square$

## A.7 ADDITIVITY AND CONNECTION TO COMPOSITIONALITY

This subsection presents a special subset of decoders that will allow autoencoders to generalize compositionally in the sense of Def. 3b.

**Definition 5b** (Additivity). A function $\boldsymbol{f} : \mathcal{Z} = \mathcal{Z}_1 \times \ldots \times \mathcal{Z}_K \to \mathbb{R}^N$ is called *additive on* $\mathcal{Z}$ if for any $k$ there exists $\boldsymbol{\varphi}_k : \mathcal{Z}_k \to \mathbb{R}^N$ such that

$$\boldsymbol{f}(\boldsymbol{z}) = \sum_{k=1}^K \boldsymbol{\varphi}_k(\boldsymbol{z}_k) \quad \text{holds for any } \boldsymbol{z} \in \mathcal{Z}. \tag{29}$$

**Lemma 7** (Compositionality implies additivity). *Let $\mathcal{Z}_k \subseteq \mathbb{R}^{KM}$ be convex sets and let $\boldsymbol{f} : \mathcal{Z} = \mathcal{Z}_1 \times \ldots \times \mathcal{Z}_K \to \mathbb{R}^N$ be $C^2$ (i.e., twice continuously differentiable) and compositional on $\mathcal{Z}$ (in the sense of Def. 4b). Then $\boldsymbol{f}$ is additive on $\mathcal{Z}$.*

*Proof of Lem. 7.* The proof is broken down into two steps. First, we prove that the coordinate functions of compositional functions have a diagonal Hessian. Second, we prove that real-valued functions defined on a convex set with diagonal Hessian are additive.

**Step 1.** Observe that $\boldsymbol{f}$ is additive if and only if for any $p \in [N]$, $\boldsymbol{f}_p$ is additive. Let $p \in [N]$ be arbitrary but fixed and let $\boldsymbol{q} := \boldsymbol{f}_p$. To be proven: $\boldsymbol{q}$ is additive. Note that since $\boldsymbol{f}$ is $C^2$, $\boldsymbol{q}$ is $C^2$ as well.

We first prove that $\boldsymbol{q}$ has a diagonal Hessian, i.e., for any $i, j \in [K], i \neq j$ we have $\partial_{ij}^2 \boldsymbol{q}(\boldsymbol{z}) = 0$ for any $\boldsymbol{z}$. Proving it indirectly, let us assume there exist $i \neq j$ and $\tilde{\boldsymbol{z}}$ such that $\partial_{ij}^2 \boldsymbol{q}(\tilde{\boldsymbol{z}}) \neq 0$.

The function $\boldsymbol{q}$ is $C^2$, thus $\partial_{ij}^2 \boldsymbol{q}$ is continuous. Therefore, there exists a neighborhood $V$ of $\tilde{\boldsymbol{z}}$ such that $\partial_{ij}^2 \boldsymbol{q}(\boldsymbol{z}) \neq 0$ for any $\boldsymbol{z} \in V$. Hence, $\partial_i \boldsymbol{q}$ cannot be constant on $V$, for otherwise $\partial_j(\partial_i \boldsymbol{q}) = \partial_{ij}^2 \boldsymbol{q}$ would be 0. Consequently, there exists $\boldsymbol{z}^* \in V$ such that $\partial_i \boldsymbol{q}(\boldsymbol{z}^*) \neq 0$. Again, $\partial_i \boldsymbol{q}$ is continuous, hence there exists a neighborhood $W \subseteq V$ of $\boldsymbol{z}^*$ such that $\partial_i \boldsymbol{q}(\boldsymbol{z}) \neq 0$ for any $\boldsymbol{z} \in W$.

However, $\boldsymbol{f}$ is compositional, hence either $\partial_i \boldsymbol{q}(\boldsymbol{z})$ or $\partial_j \boldsymbol{q}(\boldsymbol{z})$ is 0. Therefore $\partial_j \boldsymbol{q}(\boldsymbol{z}) = 0$ for any $\boldsymbol{z} \in W$. After taking the partial derivative with respect to slot $\boldsymbol{z}_i$, we receive that $\partial_{ij}^2 \boldsymbol{q}(\boldsymbol{z}) = \partial_i(\partial_j \boldsymbol{q}(\boldsymbol{z})) = 0$ for any $\boldsymbol{z} \in W \subseteq V$. This contradicts the fact that $\partial_{ij}^2 \boldsymbol{q}(\boldsymbol{z}) \neq 0$ for any $\boldsymbol{z} \in W$.

**Step 2.** Secondly, we prove that $\boldsymbol{q}$ defined on $\mathcal{Z}$ convex, having a diagonal Hessian, has to be additive. Observe that by slightly reformulating the property of a diagonal Hessian, we receive:

$$\text{for any } \boldsymbol{z}, i \text{ and } j: \quad \partial_j[\partial_i \boldsymbol{q}](\boldsymbol{z}) \neq 0 \implies j = i. \tag{30}$$

Comparing Eq. 30 to Eq. 24 from the proof of Thm. 1b, we realize that the derivative of $\mathrm{D}\boldsymbol{q}(\boldsymbol{z})$ has a blockdiagonal structure, similar to $\hat{\boldsymbol{h}}$ (except, the blocks may become 0). By repeating the same argument from Step 2. of the proof of Thm. 1b, we get that $\mathrm{D}\boldsymbol{q}$ is a slot-wise ambiguity with the identity permutation, i.e. for some $C^1$-diffeomorphisms $\boldsymbol{Q}_k$ we have $\partial_k \boldsymbol{q}(\boldsymbol{z}) = \boldsymbol{Q}_k(\boldsymbol{z}_k)$ for any $\boldsymbol{z}, k$.

However, then let $z, z^{(0)} \in \mathcal{Z}$ and let $\phi(t) : [0, 1] \to \mathcal{Z}$ be a smooth path with $\phi(0) = z^{(0)}, \phi(1) = z$ ($\mathcal{Z}$ being convex, such a path exists). Then:

$$q(z) - q(z^{(0)}) = \int_0^1 \frac{d}{dt}\big[q(\phi(t))\big]dt = \int_0^1 \mathrm{D}q(\phi(t))\,\phi'(t)dt \tag{31}$$

$$= \int_0^1 \sum_{k=1}^K \partial_k q(\phi(t))\,\phi_k'(t)dt = \sum_{k=1}^K \int_0^1 Q_k(\phi_k(t))\,\phi_k'(t)dt. \tag{32}$$

Functions $Q_k$ are continuous; hence, they can give rise to an integral function $\tilde{\varphi}_k$. Using the rule of integration by substitution, we receive:

$$q(z) - q(z^{(0)}) = \sum_{k=1}^K \tilde{\varphi}_k(\phi(t))\Big|_0^1 = \sum_{k=1}^K \big(\tilde{\varphi}_k(z_k) - \tilde{\varphi}_k(z_k^{(0)})\big). \tag{33}$$

Denoting $\varphi_k(z_k) = \tilde{\varphi}_k(z_k) - \tilde{\varphi}_k(z_k^{(0)}) + \frac{1}{K}q(z^{(0)})$, we conclude that $q$ is additive, as

$$q(z) = \sum_{k=1}^K \varphi_k(z_k). \tag{34}$$
$\square$

## A.8 Decoder Generalization

In this subsection we recall in a more precise format and prove Thm. 2. However, before that, we also precisely introduce the *slot-wise recombination space ($\mathcal{Z}'$) and -function ($h'$)*, where $\mathcal{Z}'$ is an extension of Eq. 5. The latter was only defined for the case when our autoencoder slot-identified $z$ on the training latent space.

**Definition 13** (Slot-wise recombination space and -function). Let $(\mathcal{Z}, \mathcal{Z}^S, f)$ be a POGP and let $(\hat{g}, \hat{f})$ be an autoencoder. Let $\hat{\mathcal{Z}}^S := \hat{g}(f(\mathcal{Z}^S))$. We call

$$\mathcal{Z}' := \hat{\mathcal{Z}}_1^S \times \ldots \times \hat{\mathcal{Z}}_K^S \tag{35}$$

the *slot-wise recombination space*, where $\hat{\mathcal{Z}}_k^S$ is the projection of the set $\hat{\mathcal{Z}}^S$ (c.f. Def. 8).

In the case when $(\hat{g}, \hat{f})$ slot-identifies $z$ on $\mathcal{Z}^S$ w.r.t. generator $f$, let the *slot-wise recombination function* be the concatenation of all slot-functions $h_k(z_{\pi(k)})$:

$$h'(z) := \big(h_1(z_{\pi(1)}), \ldots, h_K(z_{\pi(K)})\big). \tag{36}$$

The space of all values taken by $h'(z)$ is:

$$h'(\mathcal{Z}) = h_1(\mathcal{Z}_{\pi(1)}) \times \cdots \times h_K(\mathcal{Z}_{\pi(K)}). \tag{37}$$

Since in this case $h_k(\mathcal{Z}_{\pi(k)}) = \hat{\mathcal{Z}}_k^S$, we have that $\mathcal{Z}' = h'(\mathcal{Z})$.

**Theorem 2b** (Decoder generalization). *Let $(\mathcal{Z}, \mathcal{Z}^S, f)$ be a POGP such that*

  i) $f$ *is compositional on $\mathcal{Z}$ and*

  ii) $f$ *is $C^2$-diffeomorphism around $\mathcal{Z}$.*

*Let $(\hat{g}, \hat{f})$ be an autoencoder such that*

  iii) $(\hat{g}, \hat{f})$ *slot-identifies $z$ on $\mathcal{Z}^S$ w.r.t. $f$ and*

  iv) $\hat{f}$ *is additive (on $\mathbb{R}^{KM}$).*

*Then $\hat{f}$ generalizes in the sense that $\hat{f}(h'(z)) = f(z)$ holds for any $z \in \mathcal{Z}$. What is more, $\hat{f}$ is injective on $\mathcal{Z}'$ and we get $\hat{f}(\mathcal{Z}') = f(\mathcal{Z}) = \mathcal{X}$.*

*Proof of Thm. 2b.* Let $\mathcal{X}^S = \boldsymbol{f}(\mathcal{Z}^S)$ and $\hat{\mathcal{Z}}^S = \hat{\boldsymbol{g}}(\mathcal{X}^S)$. Condition *iii)* implies that $(\hat{\boldsymbol{g}}, \hat{\boldsymbol{f}})$ minimizes $\mathcal{L}_{\mathrm{rec}}(\mathcal{X}^S)$, or equivalently, because of Lem. 4, $\hat{\boldsymbol{g}} : \mathcal{X}^S \to \hat{\mathcal{Z}}^S$ and $\hat{\boldsymbol{f}} : \hat{\mathcal{Z}}^S \to \mathcal{X}^S$ invert each other. Also, the projection of $\hat{\mathcal{Z}}^S$ to the $k$-th slot is $\hat{\mathcal{Z}}_k^S = \boldsymbol{h}_k(\mathcal{Z}_{\pi(k)})$. Furthermore, from Def. 13 we know that $\hat{\boldsymbol{g}}\big(\boldsymbol{f}(\boldsymbol{z})\big) = \boldsymbol{h}'(\boldsymbol{z})$, for any $\boldsymbol{z} \in \mathcal{Z}^S$. Hence, by applying $\hat{\boldsymbol{f}}$ on both sides, we receive:

$$\boldsymbol{f}(\boldsymbol{z}) = \hat{\boldsymbol{f}}\big(\boldsymbol{h}'(\boldsymbol{z})\big) \text{ for any } \boldsymbol{z} \in \mathcal{Z}^S. \tag{38}$$

Besides, $\hat{\boldsymbol{f}}$ is additive. Due to *ii)* and Lem. 7, $\boldsymbol{f}$ is also additive on $\mathcal{Z}$. More precisely: for any $k \in [K]$ there exist functions $\boldsymbol{\varphi}_k : \mathcal{Z}_k \to \mathbb{R}^N, \hat{\boldsymbol{\varphi}}_k : \boldsymbol{h}_k(\mathcal{Z}_{\pi(k)}) \to \mathbb{R}^N$ such that:

$$\boldsymbol{f}(\boldsymbol{z}) = \sum_{k=1}^K \boldsymbol{\varphi}_k(\boldsymbol{z}_k) \text{ for any } \boldsymbol{z} \in \mathcal{Z} \text{ and} \tag{39}$$

$$\hat{\boldsymbol{f}}(\hat{\boldsymbol{z}}) = \sum_{k=1}^K \hat{\boldsymbol{\varphi}}_k(\hat{\boldsymbol{z}}_k) \text{ for any } \hat{\boldsymbol{z}} \in \mathcal{Z}'. \tag{40}$$

Substituting this into Eq. 38, we receive:

$$\sum_{k=1}^K \boldsymbol{\varphi}_k(\boldsymbol{z}_k) = \sum_{k=1}^K \hat{\boldsymbol{\varphi}}_k\big(\boldsymbol{h'}_k(\boldsymbol{z})\big) = \sum_{k=1}^K \hat{\boldsymbol{\varphi}}_k\big(\boldsymbol{h}_k(\boldsymbol{z}_{\pi(k)})\big) \text{ for any } \boldsymbol{z} \in \mathcal{Z}^S. \tag{41}$$

It is easily seen that functions $\boldsymbol{\varphi}_k, \hat{\boldsymbol{\varphi}}_k$ are $C^1$. After differentiating Eq. 41 with respect to $\boldsymbol{z}_k$, we receive:

$$\mathrm{D}\boldsymbol{\varphi}_k(\boldsymbol{z}_k) = \mathrm{D}\left(\hat{\boldsymbol{\varphi}}_{\pi^{-1}(k)} \circ \boldsymbol{h}_{\pi^{-1}(k)}\right)(\boldsymbol{z}_k) \text{ for any } \boldsymbol{z} \in \mathcal{Z}^S. \tag{42}$$

Since $\mathcal{Z}^S$ is a slot-supported subset of $\mathcal{Z}$, Eq. 42 holds for any $\boldsymbol{z}_k \in \mathcal{Z}_k$. Let us denote $\gamma_k = \hat{\boldsymbol{\varphi}}_{\pi^{-1}(k)} \circ \boldsymbol{h}_{\pi^{-1}(k)} \in C^1(\mathcal{Z}_k)$.

Since $\mathcal{Z}_k$ is convex, let $\boldsymbol{z}_k^{(0)} \in \mathcal{Z}_k$ fixed and define the path $t \in [0, 1] \mapsto u(t) = \boldsymbol{z}_k^{(0)} + t(\boldsymbol{z}_k - \boldsymbol{z}_k^{(0)}) \in \mathcal{Z}_k$. Then:

$$\boldsymbol{\varphi}_k(\boldsymbol{z}_k) - \boldsymbol{\varphi}_k(\boldsymbol{z}_k^{(0)}) = \boldsymbol{\varphi}_k\big(u(t)\big)\Big|_0^1 = \int_0^1 \mathrm{D}\boldsymbol{\varphi}_k(u(t))\, u'(t)dt \tag{43}$$

Due to Eq. 42, we may continue:

$$\boldsymbol{\varphi}_k(\boldsymbol{z}_k) - \boldsymbol{\varphi}_k(\boldsymbol{z}_k^{(0)}) = \int_0^1 \mathrm{D}\gamma_k(u(t))\, u'(t)dt = \gamma_k\big(u(t)\big)\Big|_0^1 = \gamma_k(\boldsymbol{z}_k) - \gamma_k(\boldsymbol{z}_k^{(0)}). \tag{44}$$

Consequently, there exist constants $\boldsymbol{c}_k \in \mathbb{R}^N$ such that for any $k$:

$$\boldsymbol{\varphi}_k(\boldsymbol{z}_k) = \hat{\boldsymbol{\varphi}}_{\pi^{-1}(k)}\big(\boldsymbol{h}_{\pi^{-1}(k)}(\boldsymbol{z}_k)\big) + \boldsymbol{c}_k \quad \text{for any } \boldsymbol{z}_k \in \mathcal{Z}_k. \tag{45}$$

After adding them up for all $k$ and using Eq. 39:

$$\boldsymbol{f}(\boldsymbol{z}) = \sum_{k=1}^K \boldsymbol{\varphi}_k(\boldsymbol{z}_k) = \sum_{k=1}^K \left(\hat{\boldsymbol{\varphi}}_{\pi^{-1}(k)}\big(\boldsymbol{h}_{\pi^{-1}(k)}(\boldsymbol{z}_k)\big) + \boldsymbol{c}_k\right) \tag{46}$$

$$= \sum_{k=1}^K \hat{\boldsymbol{\varphi}}_k\big(\boldsymbol{h}_k(\boldsymbol{z}_{\pi(k)})\big) + \sum_{k=1}^K \boldsymbol{c}_k. \tag{47}$$

Now, based on Eq. 40, we receive:

$$\boldsymbol{f}(\boldsymbol{z}) = \hat{\boldsymbol{f}}\big(\boldsymbol{h}_1(\boldsymbol{z}_{\pi(1)}), \ldots, \boldsymbol{h}_K(\boldsymbol{z}_{\pi(K)})\big) + \sum_{k=1}^K \boldsymbol{c}_k = \hat{\boldsymbol{f}}\big(\boldsymbol{h}'(\boldsymbol{z})\big) + \sum_{k=1}^K \boldsymbol{c}_k \tag{48}$$

holds for any $\boldsymbol{z} \in \mathcal{Z}$. In particular, Eq. 48 holds for $\boldsymbol{z} \in \mathcal{Z}^S$, which together with 38 implies that

$$\sum_{k=1}^K \boldsymbol{c}_k = 0.$$

Finally, we arrive at:

$$\boldsymbol{f}(\boldsymbol{z}) = \hat{\boldsymbol{f}}\big(\boldsymbol{h}'(\boldsymbol{z})\big) \text{ for any } \boldsymbol{z} \in \mathcal{Z}. \tag{49}$$

Note that previously Eq. 38 only held for $\boldsymbol{z} \in \mathcal{Z}^S$.

Moreover, since $\boldsymbol{h}' : \mathcal{Z} \to \hat{\mathcal{Z}}$ is a diffeomorphism, we see that

$$\boldsymbol{f}(\mathcal{Z}) = \hat{\boldsymbol{f}}\big(\boldsymbol{h}'(\mathcal{Z})\big) = \hat{\boldsymbol{f}}(\mathcal{Z}'). \tag{50}$$

□

**Remark.** In the process of the proof, we have also proven the identifiability of the slot-wise additive components up to constant translations: There exists a permutation $\pi \in \mathcal{S}(K)$ and for any $k$, constants $\boldsymbol{c}_k$ such that

$$\boldsymbol{\varphi}_k(\boldsymbol{z}_k) = \hat{\boldsymbol{\varphi}}_{\pi^{-1}(k)}\big(\boldsymbol{h}_{\pi^{-1}(k)}(\boldsymbol{z}_k)\big) + \boldsymbol{c}_k \quad \text{for any } \boldsymbol{z}_k \in \mathcal{Z}_k. \tag{51}$$

### A.9 Encoder Generalization

**Definition 6b** (Compositional consistency)**.** Let $\big(\hat{\boldsymbol{g}}, \hat{\boldsymbol{f}}\big)$ be an autoencoder. Let $\mathcal{Z}' \subseteq \mathbb{R}^{KM}$ be an arbitrary set and let $q_{\boldsymbol{z}'}$ be an arbitrary continuous distribution with $\mathrm{supp}(q_{\boldsymbol{z}'}) = \mathcal{Z}'$. The following quantity is called the *compositional consistency loss of* $\big(\hat{\boldsymbol{g}}, \hat{\boldsymbol{f}}\big)$ *with respect to* $q_{\boldsymbol{z}'}$:

$$\mathcal{L}_{\mathrm{cons}}\big(\hat{\boldsymbol{g}}, \hat{\boldsymbol{f}}, q_{\boldsymbol{z}'}\big) = \mathbb{E}_{\boldsymbol{z}' \sim q_{\boldsymbol{z}'}}\Big[\big\|\hat{\boldsymbol{g}}\big(\hat{\boldsymbol{f}}(\boldsymbol{z}')\big) - \boldsymbol{z}'\big\|_2^2\Big]. \tag{52}$$

We say that $\big(\hat{\boldsymbol{g}}, \hat{\boldsymbol{f}}\big)$ is *compositionally consistent* if the *compositional consistency loss* w.r.t. $q_{\boldsymbol{z}'}$ vanishes.

We now state a lemma that, similarly to Lem. 4, states that for $\mathcal{L}_{\mathrm{cons}}\big(\hat{\boldsymbol{g}}, \hat{\boldsymbol{f}}, q_{\boldsymbol{z}'}\big)$ to vanish, the exact choice of $q_{\boldsymbol{z}'}$ is irrelevant and that in this case the decoder (right) inverts the encoder on $\mathcal{Z}'$, meaning that the autoencoder is consistent on $\hat{\boldsymbol{f}}(\mathcal{Z}')$ to $\mathcal{Z}'$.

**Lemma 8** (Vanishing $\mathcal{L}_{\mathrm{cons}}$ implies invertibility on $\mathcal{Z}'$)**.** *For* $\big(\hat{\boldsymbol{g}}, \hat{\boldsymbol{f}}\big)$ *and* $\mathcal{Z}' \subseteq \mathbb{R}^{KM}$ *the following statements are equivalent:*

  i) *there exists* $q_{\boldsymbol{z}'}$ *with* $\mathrm{supp}(q_{\boldsymbol{z}'}) = \mathcal{Z}'$ *such that* $\mathcal{L}_{cons}\big(\hat{\boldsymbol{g}}, \hat{\boldsymbol{f}}, q_{\boldsymbol{z}'}\big) = 0$*;*

 ii) *for any* $q_{\boldsymbol{z}'}$ *with* $\mathrm{supp}(p_{\boldsymbol{x}}) = \mathcal{Z}'$ *we have* $\mathcal{L}_{cons}\big(\hat{\boldsymbol{g}}, \hat{\boldsymbol{f}}, q_{\boldsymbol{z}'}\big) = 0$*;*

iii) $\hat{\boldsymbol{g}} \circ \hat{\boldsymbol{f}}|_{\mathcal{Z}'} = id$ *or, in other words,* $\big(\hat{\boldsymbol{g}}, \hat{\boldsymbol{f}}\big)$ *is consistent on* $\hat{\boldsymbol{f}}(\mathcal{Z}')$ *to* $\mathcal{Z}'$*.*

**Remark.** The proof is analogous to the one of Lem. 4. Henceforth, in the context of vanishing consistency loss, we are going to denote $\mathcal{L}_{\mathrm{cons}}\big(\hat{\boldsymbol{g}}, \hat{\boldsymbol{f}}, q_{\boldsymbol{z}'}\big)$ simply either by $\mathcal{L}_{\mathrm{cons}}\big(\hat{\boldsymbol{g}}, \hat{\boldsymbol{f}}, \mathcal{Z}'\big)$ or $\mathcal{L}_{\mathrm{cons}}(\mathcal{Z}')$.

**Theorem 3b** (Compositionally generalizing autoencoder)**.** *Let* $(\mathcal{Z}, \mathcal{Z}^S, \boldsymbol{f})$ *be a POGP (Def. 9) such that*

  i) $\mathcal{Z}^S$ *is convex,*

 ii) $\boldsymbol{f}$ *is compositional and irreducible on* $\mathcal{Z}$ *and*

iii) $\boldsymbol{f}$ *is* $C^2$*-diffeomorphism around* $\mathcal{Z}$*.*

*Let* $\big(\hat{\boldsymbol{g}}, \hat{\boldsymbol{f}}\big)$ *be an autoencoder with* $\mathcal{X}^S = \boldsymbol{f}(\mathcal{Z}^S)$ *(Def. 10) such that*

 iv) $\big(\hat{\boldsymbol{g}}, \hat{\boldsymbol{f}}\big)$ *minimizes* $\mathcal{L}_{rec}\big(\hat{\boldsymbol{g}}, \hat{\boldsymbol{f}}, \mathcal{X}^S\big) + \lambda \mathcal{L}_{cons}\big(\hat{\boldsymbol{g}}, \hat{\boldsymbol{f}}, \mathcal{Z}'\big)$ *for some* $\lambda > 0$*, where* $\mathcal{Z}'$ *is the slot-wise recombination space (see definition of* $\mathcal{Z}'$ *in Def. 13),*

  v) $\hat{\boldsymbol{f}}$ *is compositional on* $\hat{\mathcal{Z}}^S := \hat{\boldsymbol{g}}(\mathcal{X}^S)$ *and*

 vi) $\hat{\boldsymbol{f}}$ *is additive (on* $\mathbb{R}^{KM}$*).*

*Then the autoencoder* $\big(\hat{\boldsymbol{g}}, \hat{\boldsymbol{f}}\big)$ *generalizes compositionally w.r.t.* $\mathcal{Z}^S$ *in the sense of Def. 3b. Moreover,* $\hat{\boldsymbol{g}} : \mathcal{X} \to \hat{\mathcal{Z}} = \hat{\boldsymbol{g}}(\mathcal{X})$ *inverts* $\hat{\boldsymbol{f}} : \mathcal{Z}' \to \mathcal{X}$ *and* $\hat{\mathcal{Z}} = \mathcal{Z}' = \boldsymbol{h}_1(\mathcal{Z}_{\pi(1)}) \times \cdots \times \boldsymbol{h}_K(\mathcal{Z}_{\pi(K)})$*.*

*Proof of Thm. 3b.* Observe that assumption *iv)* is equivalent to

$$\mathcal{L}_{\text{rec}}(\hat{\boldsymbol{g}}, \hat{\boldsymbol{f}}, \mathcal{X}^S) + \lambda \mathcal{L}_{\text{cons}}(\hat{\boldsymbol{g}}, \hat{\boldsymbol{f}}, \mathcal{Z}') = 0, \tag{53}$$

which may happen if and only if $\mathcal{L}_{\text{rec}}(\hat{\boldsymbol{g}}, \hat{\boldsymbol{f}}, \mathcal{X}^S) = \mathcal{L}_{\text{cons}}(\hat{\boldsymbol{g}}, \hat{\boldsymbol{f}}, \mathcal{Z}') = 0$, i.e. $(\hat{\boldsymbol{g}}, \hat{\boldsymbol{f}})$ minimizes both $\mathcal{L}_{\text{rec}}(\mathcal{X}^S)$ and $\mathcal{L}_{\text{cons}}(\mathcal{Z}')$.

Firstly, as *i)* $\boldsymbol{f}$ is compositional on $\mathcal{Z}$, and hence on $\mathcal{Z}^S$, *ii)* $\hat{\boldsymbol{f}}$ is compositional on $\hat{\mathcal{Z}}^S := \hat{\boldsymbol{g}}(\mathcal{X}^S)$ and *iii)* $(\hat{\boldsymbol{g}}, \hat{\boldsymbol{f}})$ minimizes $\mathcal{L}_{\text{rec}}(\mathcal{X}^S)$, we may conclude based on Thm. 1b that $(\hat{\boldsymbol{g}}, \hat{\boldsymbol{f}})$ slot-identifies $\boldsymbol{z}$ on $\mathcal{Z}^S$. Consequently, the slot-wise recombination function $\boldsymbol{h}'$ is well-defined.

Secondly, *i)* $\boldsymbol{f}$ is compositional on $\mathcal{Z}$; *ii)* $\boldsymbol{f} \in C^2 \text{Diffeo}(\mathcal{Z})$; *iii)* $\hat{\boldsymbol{f}}$ is additive and *iv)* $(\hat{\boldsymbol{g}}, \hat{\boldsymbol{f}})$ slot-identifies $\boldsymbol{z}$ on $\mathcal{Z}^S$. Therefore, Thm. 2b implies that $\hat{\boldsymbol{f}}$ in injective on $\mathcal{Z}'$, $\hat{\boldsymbol{f}}(\boldsymbol{h}'(\boldsymbol{z})) = \boldsymbol{f}(\boldsymbol{z})$ holds for any $\boldsymbol{z} \in \mathcal{Z}$ and $\hat{\boldsymbol{f}}(\mathcal{Z}') = \boldsymbol{f}(\mathcal{Z}) = \mathcal{X}$.

Finally, from Lem. 8 and the fact that $(\hat{\boldsymbol{g}}, \hat{\boldsymbol{f}})$ minimizes $\mathcal{L}_{\text{cons}}(\mathcal{Z}')$, we deduce that $(\hat{\boldsymbol{g}}, \hat{\boldsymbol{f}})$ is consistent on $\hat{\boldsymbol{f}}(\mathcal{Z}')$, meaning that $\hat{\boldsymbol{g}}$ is injective on $\hat{\boldsymbol{f}}(\mathcal{Z}') = \mathcal{X}$ and its inverse is $\hat{\boldsymbol{f}} : \mathcal{Z}' \to \mathcal{X}$. However, by definition, $\hat{\boldsymbol{g}}(\mathcal{X}) = \hat{\mathcal{Z}}$. Consequently, we proved that $\hat{\mathcal{Z}} = \mathcal{Z}'$.

Furthermore, we recall that $\hat{\boldsymbol{f}}(\boldsymbol{h}'(\boldsymbol{z})) = \boldsymbol{f}(\boldsymbol{z})$ holds for any $\boldsymbol{z} \in \mathcal{Z}$. As $\hat{\boldsymbol{f}}$ is invertible on $\mathcal{Z}' = \hat{\mathcal{Z}}$ with inverse $\hat{\boldsymbol{g}}$, we may pre-apply $\hat{\boldsymbol{g}}$ on both sides and receive

$$\hat{\boldsymbol{g}}(\boldsymbol{f}(\boldsymbol{z})) = \boldsymbol{h}'(\boldsymbol{z}), \quad \text{for any } \boldsymbol{z} \in \mathcal{Z}, \tag{54}$$

which proves that $(\hat{\boldsymbol{g}}, \hat{\boldsymbol{f}})$ also slot-identifies $\boldsymbol{z}$ on $\mathcal{Z}$ and concludes our proof. □

## B  EXPERIMENT DETAILS

### B.1  DATA GENERATION

The multi-object sprites dataset used in all experiments was generated using DeepMind's Sprite-world renderer (Watters et al., 2019). Each image consists of two sprites where the ground-truth latents for each sprite were sampled uniformly in the following intervals: x-position in $[0.1, 0.9]$, y-position in $[0.2, 0.8]$, shape in $\{0, 1\}$ corresponding to $\{$triangle, square$\}$, scale in $[0.09, 0.22]$, and color (HSV) in $[0.05, 0.95]$ where saturation and value are fixed and only hue is sampled.

All latents were scaled such that their sampling intervals become equivalent to a hypercube $[0, 1]^{2 \times 5}$ (2 slots with 5 latents each) and then scaled back to their original values before rendering. The slot-supported subset $\mathcal{Z}^S$ of the latent space was defined as a slot-wise band around the diagonal through this hypercube with width $\delta = 0.25$ along each latent slot, i.e.,

$$\mathcal{Z}^S = \left\{ (\boldsymbol{z}_1, \boldsymbol{z}_2) | \forall i \in [5] : (\boldsymbol{z}_1 - \boldsymbol{z}_2)_i \leq \sqrt{2}\delta \right\}. \tag{55}$$

In this sampling region, sprites would almost entirely overlap for small $\delta$. Therefore, we apply an offset to the x-position latent of slot $\boldsymbol{z}_2$. Specifically, we set it to $(x + 0.5) \mod 1$, where $x$ is the sampled x-position.

The training set and ID test set were then sampled uniformly from the resulting region, $\mathcal{Z}^S$, while the OOD test set was sampled uniformly from $\mathcal{Z} \setminus \mathcal{Z}^S$. Objects with Euclidean distance smaller than 0.2 in their position latents were filtered to avoid overlaps. The resulting training set consists of 100,000 samples, while the ID and OOD test set each consist of 5,000 samples. Each rendered image is of size $64 \times 64 \times 3$.

### B.2  MODEL ARCHITECTURE AND TRAINING SETTINGS

The encoder and decoder for the additive autoencoder used in Sec. 6.1 closely resemble the architectures from Burgess et al. (2018). The encoder consists of four convolutional layers, followed by four linear layers, and outputs a vector of dimensions $2 \times h$, where $h$ represents the latent size of a single slot and is set to 16. The decoder consists of three linear layers, followed by four transposed convolutional layers, and is applied to each slot separately; the slot-wise reconstructions are

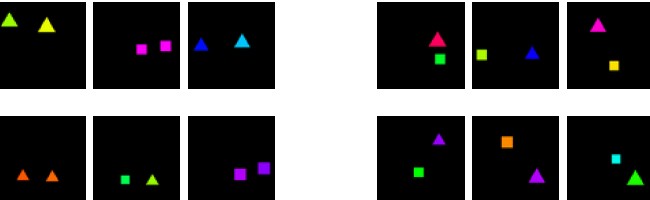

Figure 6: **Samples from dataset used in Sec. 6**. **Left**: In-distribution samples with latents sampled from the diagonal region. Objects are highly correlated in all latents and use an offset in their x-position to avoid direct overlaps. **Right**: Out-of-distribution samples with latents sampled from the off-diagonal region.

subsequently summed to produce the final output. The ReLU activations were replaced with ELU in both the encoder and decoder. The Slot Attention model in Sec. 6.2 follows the implementation from the original paper Locatello et al. (2020a), where hyperparameters were chosen in accordance with the Object-Centric library (Dittadi et al., 2021) for the Multi-dSprites setting, but with the slot dimension set to 16.

The additive autoencoder is trained for 300 epochs, while Slot Attention is trained for 400 epochs. Both models are trained using a batch size of 64. We optimize both models with AdamW (Loshchilov and Hutter, 2019) with a warmup of eleven epochs. The initial learning rate is set as $1 \times 10^{-7}$ and doubles every epoch until it reaches the value of $0.0004$. Subsequently, the learning rate is halved every 50 epochs until it reaches $1 \times 10^{-7}$. Both models were trained using PyTorch (Paszke et al., 2019).

For the experiments verifying our theoretical results in Sec. 6.1, we only included models in our results which were able to adequately minimize their training objective. More specifically, we only selected models that achieved reconstruction loss on the ID test set less than $2.0$. For the experiments with Slot Attention in Sec. 6.2, we only reported results for models that were able to separate objects ID where seeds were selected by visual inspection. This was done since the primary purpose of these experiments was not to see the effect of our assumptions on slot identifiability ID but instead OOD. Thus, we aimed to ensure that the effect of our theoretical assumptions on our OOD metrics were not influenced by the confounder of poor slot identifiability ID. Using these selection criteria gave us between 5 and 10 seeds for both models, which were used to compute our results.

If used, the composition consistency loss is introduced with $\lambda = 1$ from epoch 100 onwards for the additive autoencoder model and epoch 150 onwards for Slot Attention. The number of recombined samples $z'$ in each forward pass of the consistency loss is equal to the batch size for all experiments. In our compositional consistency implementation, normalization of both the latents $z'$ and the re-encoded latents $\hat{g}(\hat{f}(z'))$ proved to be essential before matching them with the Hungarian algorithm and calculating the loss value. Without this normalization, we encountered numerical instabilities, which resulted in exploding gradients.

### B.3 MEASURING RECONSTRUCTION ERROR

For the heatmaps in Figs. 1 and 5, we first calculate the normalized reconstruction MSE on both the ID and OOD test sets. We then project the ground-truth latents of each test point onto a 2D plane with the x and y-axes corresponding to the color latent of each object. We report the average MSE in each bin. In this projection, some OOD points would end up in the ID region. Since we do not observe a difference in MSE for OOD points that are projected to the ID or OOD region, we simply filter those OOD points to allow for a clear visual separation of the regions.

When reporting the isolated decoder reconstruction error in Fig. 1 A and Fig. 5 A and B, we aim to visualize how much of the overall MSE can attributed to only the decoder. Since the models approximately slot-identify the ground-truth latents ID on the training distribution, the MSE of the full autoencoder serves as a tight upper bound for the isolated decoder reconstruction error. Thus, in the ID region, we use this MSE when reporting the isolated decoder error. For the OOD region,

however, the reconstruction error could be attributed to a failure of the encoder to infer the correct latent representations or to a failure of the decoder in rendering the inferred latents. To remove the effect of the encoder's OOD generalization error, we do the following: For a given OOD test image, we find two ID test images that each contain one of the objects in the OOD image in the same configuration. Because the encoder is approximately slot identifiable ID, we know that the correct representation OOD for each object in the image is given by the encoder's ID representation for the individual objects in both ID images. To get these ID representations, we must solve a matching problem to find which ID latent slot corresponds to a given object in each image. We do this by matching slot-wise renders of the decoder with the ground-truth slot-wise renders for each object based on MSE using the Hungarian algorithm (Kuhn, 1955). Using this representation then allows us to obtain the correct representation for an OOD image without relying on the encoder to generalize OOD. The entire reconstruction error on this image can thus be attributed to a failure of the decoder to generalize OOD.

### B.4 COMPOSITIONAL CONTRAST

The compositional contrast given by Brady et al. (2023) is defined as follows:

**Definition 14** (Compositional Contrast). Let $\boldsymbol{f} : \mathcal{Z} \to \mathcal{X}$ be differentiable. The *compositional contrast* of $\boldsymbol{f}$ at $\boldsymbol{z}$ is

$$C_{\text{comp}}(\boldsymbol{f}, \boldsymbol{z}) = \sum_{n=1}^{N} \sum_{k=1}^{K} \sum_{j=k+1}^{K} \left\| \frac{\partial f_n}{\partial \boldsymbol{z}_k}(\boldsymbol{z}) \right\| \left\| \frac{\partial f_n}{\partial \boldsymbol{z}_j}(\boldsymbol{z}) \right\|. \tag{56}$$

This contrast function was proven to be zero if and only if $\boldsymbol{f}$ is compositional according to Def. 4. The function can be understood as computing each pairwise product of the (L2) norms for each pixel's gradients w.r.t. any two distinct slots $k \neq j$ and taking the sum. This quantity is non-negative and will only be zero if each pixel is affected by at most one slot such that $\boldsymbol{f}$ satisfies Def. 4. We use this contrast function to measure compositionality of a decoder in our experiments in Sec. 6.1. More empirical and theoretical details on the function may be found in Brady et al. (2023).

### B.5 COMPUTATIONAL COST OF THE PROPOSED CONSISTENCY LOSS

Computing the consistency loss as illustrated in Fig. 3 poses some additional computational overhead. Namely, it requires additional passes through the encoder and decoder as well as computation of the encoder's gradients w.r.t. the loss. As outlined in Sec. 4, we computed the consistency loss on samples $\boldsymbol{z}'$ obtained by randomly shuffling the slots in the current batch, effectively doubling the batch size. We found this to increase training time by a maximum of 28 % across runs.

For two slots, it would be possible to sample up to $b(b-1)$ novel combinations of the slots within the given batch, where $b$ is the batch size. This number increases combinatorially to $\frac{b!}{(b-n)!}$ for $n$ slots, which could pose a severe computational overhead. While our experiments demonstrate that the loss works well with just $b$ samples, App. C.3 also shows that it does not scale well to more than two slots, and drawing more samples might be a way to remedy this.

On the other hand, there might be ways to draw samples more effectively. One approach could be to sample slot combinations in proportion to their value w.r.t. the consistency loss or to sample combinations according to their likelihood under a prior over latents. Using a likelihood could avoid sampling implausible combinations; however, such a scheme is challenging as it relies on the likelihood being valid for OOD combinations of slots. Another possibility would be to include heuristics to directly filter combinations based on a priori knowledge of the data-generation process, e.g., to filter objects with similar coordinates which would intersect.

### B.6 USING MORE REALISTIC DATASETS

Extending our experiments from the sprites dataset to more realistic data poses two main challenges. Firstly, in real-world datasets one generally does not have access to the ground-truth latents making our evaluation schemes inapplicable. Secondly, even if access to ground-truth latent information is available, our experiments require being able to sample latents densely from a slot-supported subset.

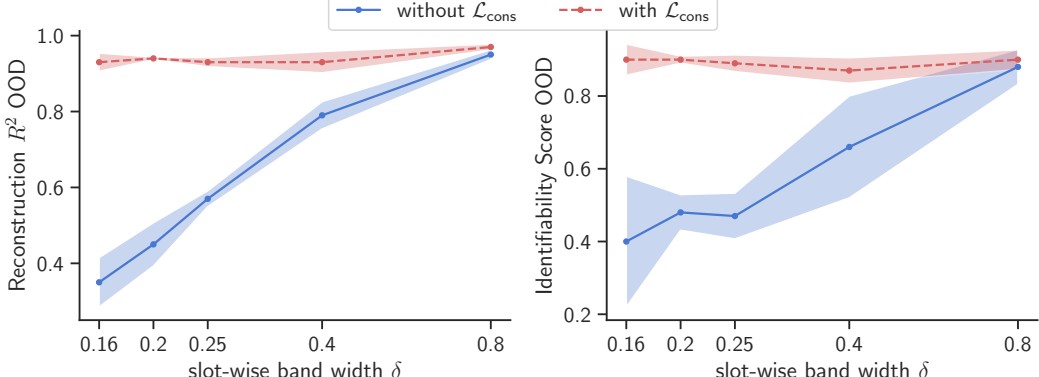

Figure 7: **OOD Reconstruction quality and slot identifiability as a function of training region size**. The width $\delta \in (0, 1)$ of the slot-wise band around the diagonal (see Eq. 55) is 0 if $\mathcal{Z}^S$ is a line and 1 if $\mathcal{Z}^S = \mathcal{Z}$; experiments in Sec. 6 used $\delta = 0.25$. In the absence of the consistency loss (Def. 6), a large $\delta$ is required for models to achieve high OOD performance for reconstruction and slot identifiability. In contrast, models trained with the consistency loss yield consistently high performance across all $\delta$. Results are averaged over at least four random seeds.

Specifically, our experiments rely on sampling from a diagonal strip in the latent space with small width. If such a region were sub-sampled from an existing dataset, this would leave only a very small number of data points which are insufficient to train a model. To this end, our experiments require access to the ground-truth renderer for a dataset such that latents can be sampled densely. This is not available in most cases, however. An interesting avenue to address this would be to leverage recent rendering pipelines such as Kubric (Greff et al., 2022) to create more complex synthetic datasets. We leave this as an interesting direction for future work.

## C  ADDITIONAL EXPERIMENTS AND FIGURES

This section provides additional experimental results to the main experiments from Sec. 6.

### C.1  IMPACT OF TRAINING REGION SIZE

We ablate the impact of the size of slot-supported subset on OOD metrics in Fig. 7 by varying the width $\delta$ of the slot-wise band around the diagonal (see Eq. 55). All models use an additive decoder and differ in whether they optimize the consistency loss (Def. 6). We see that models which do not optimize the loss require an increasingly large $\delta$ in order to achieve high OOD reconstruction and identifiability scores, while models which do optimize the loss, achieve consistently high scores on OOD metrics across all values of $\delta$.

### C.2  VIOLATING SLOT-SUPPORTEDNESS

We illustrate the effect of violating the assumption that $\mathcal{Z}^S$ is a *slot-supported* subset of $\mathcal{Z}$ (recall Def. 1) in Fig. 8 on reconstruction loss for an additive autoencoder trained with consistency loss. To do this, we create a gap in the slot-supported subset $\mathcal{Z}^S$ by removing all occurrences of objects with a hue-latent in the interval $(0.5, 0.8)$. We can see in Fig. 8 that this leads to poor reconstruction performance in the region containing the gap which propagates to the OOD regions as well.

### C.3  IMPACT OF MORE THAN 2 OBJECTS ON CONSISTENCY LOSS

We also examine how the consistency loss scales as the number of objects in the training data is increased from two to three and four. We find that as the number of objects grows, optimization of the consistency loss becomes more challenging (Fig. 9, bottom left) which makes sense considering that the number of possible slot combinations grows combinatorially with the number of slots. This,

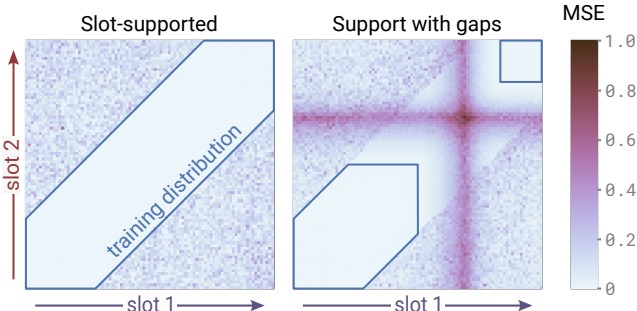

Figure 8: **Violating slot-supportedness prevents OOD generalization**. We retrain the additive autoencoder from Sec. 6.1 on data arising from a latent subset $\mathcal{Z}^S$ in which all occurrences of objects with a hue-latent in the interval $(0.5, 0.8)$ have been removed. This leads to a cross-shaped gap in the latent subset $\mathcal{Z}^S$ which can be visualized via a 2D projection of the latent space (see App. B.3 for details) where the x- and y-axes correspond to the hue-latent of each object (right). Compared to a model trained without this gap (left), we can see that the model is unable to reconstruct ID samples in this gap (i.e., samples where either object has this hue), and this error propagates outward to OOD samples.

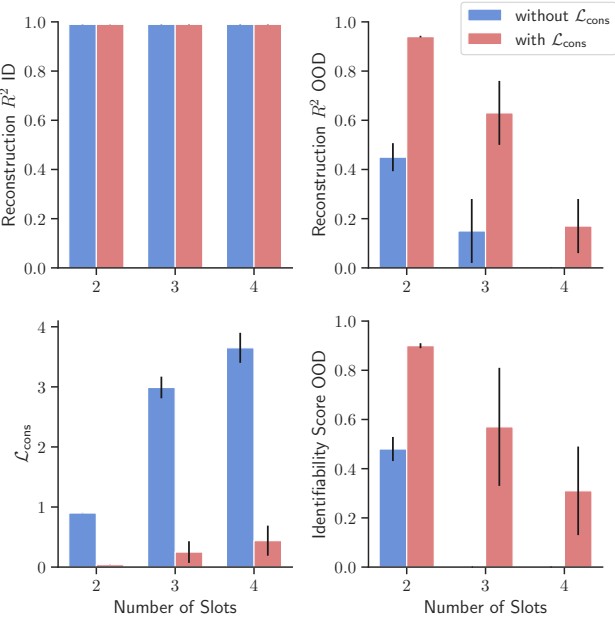

Figure 9: **Impact of number of objects on consistency loss**. We measure how the consistency loss scale as the number of objects in the training data is increased from two to three and four. We measure this for additive autoencoders which explicitly optimize the consistency loss (red) and models which do not optimize it (blue). **Top and Bottom left**: We can see that the ID reconstruction remains high as the number of objects grow, but the consistency loss increases steeply across models. **Top right**: Consequently, the OOD reconstruction quality decreases as the number of objects increases. **Bottom right**: This then prevents the encoder from slot-identifying the ground-truth latents OOD. However, training with the consistency loss still yields generally better results than training without it. All results are averaged over at least four random seeds. The consistency loss is normalized by the number of slots, and $R^2$ scores are clipped to zero.

in turn, leads to poor OOD reconstruction quality (Fig. 9, top right) which prevents the encoder from slot-identifying the ground-truth latents (Fig. 9, bottom right). As hypothesized in App. B.5, more principled schemes for sampling slot combinations could mitigate these scaling issues.

