# OpenReview forum: "Provable Compositional Generalization for Object-Centric Learning"
_ICLR.cc/2024/Conference — ICLR 2024 oral_

### Official Review · Reviewer_JA2G · 2023-10-29

**Soundness:** 4 excellent
**Presentation:** 4 excellent
**Contribution:** 3 good
**Rating:** 8
**Confidence:** 4

**Summary:**

The authors theoretically and empirically show that compositional generalization can be achieved through:
1. Structural constraints on the decoder (each data dimension is rendered as the sum of functions operating on slots separately), which ensures that the decoder compositionally generalizes.
2. An encoder-decoder consistency loss (reconstruction loss for the representations) on slot-shuffled representations from the encoder output, which encourages the encoder to compositionally generalize with the additive decoder.

The paper provides a joint encoder-decoder framework for compositional generalization for autoencoders, where previous work has mostly focused on specific aspects of the setting.

**Strengths:**

- The paper is very well-written.
- The theory is sound and significant for the community.
- The joint encoder-decoder framework for compositional generalization in autoencoders is quite elegant.
- The limitations of the framework and the additivity constraint on the decoder are adequately stated.

**Weaknesses:**

Although they support the theory, the experiments are quite limited. For instance, these are all with only two slots with 16 dimensions each. See the questions section for additional information that would be interesting to see from experimentation.

**Questions:**

- How does the effect of the consistency loss scale with the number of slots?
- What is the impact of how slot-supported the training data is? i.e. in Figure 2 (1), what is the impact of the width of the blue band on empirical effectivity?
- How does the method hold up on non-synthetic data, especially if you slightly relax some constraints? For instance, what if you have expressive slot-wise decoding, but allow for a low-expressivity non-linear combination at the end for rendering?

---

> ### Author Response · Authors · 2023-11-21
>
> Dear Reviewer,
>
> We thank you for your positive review and your valuable feedback. We were delighted to see that you found our theory “sound and significant”, that you found our proposed framework “elegant”, and that you found our manuscript to be “very well written”.
>
> We address your comments below.
>
> ___
> **Comment:** ”How does the effect of the consistency loss scale with the number of slots?”
>
> **Response**:
> Thank you for asking this interesting question! In App. C.3, we now include an experiment that ablates the impact of the number of slots on the consistency loss. As the number of slots increases from 2 to 3 and 4, optimization of the consistency loss becomes more challenging, and its value grows. This is not unexpected as the number of possible latent slot combinations grows combinatorially with the number of slots, making sampling the space more challenging. Furthermore, as the consistency loss increases, we see worse performance on OOD metrics, as our theory suggests. We now include a discussion of potential solutions for scaling this loss to more complex settings in App. B.5.
>
> ___
> **Comment:** “What is the impact of how slot-supported the training data is? What is the impact of the width of the blue band on empirical effectivity?"
>
> **Response**:
> Thank you for raising this interesting point! We address this question by examining two questions empirically:
>
> 1. What is the effect of the width of the diagonal band in latent space, i.e., the effect of the training data's size (and diversity)?
> 2. What happens if slot-supportedness is not fulfilled?
>
> We address the first point in App. C.1 and show that a model trained with the consistency loss yields high OOD performance across different widths. On the other hand, a model trained without consistency loss requires a much wider band to reach the same OOD performance.
>
> We address the second point in App. C.2 by creating a dataset in which there is a gap in the latent space such that it is no longer a slot-supported subset. In this case, we see that OOD reconstruction increases sharply in the region around this gap.
>
> ___
> **Comment:** ”How does the method hold up on non-synthetic data?”
>
> **Response**:
> We agree with the reviewer that understanding our results on non-synthetic data is important; however, this is challenging for two main reasons. Firstly, for real-world data, one does not generally have access to the ground-truth latents making our evaluation schemes inapplicable. Secondly, even if access to ground-truth latent information is available, our experiments require being able to sample latents densely from a slot-supported subset. Specifically, our experiments rely on sampling from a diagonal strip in the latent space with a small width. If such a region were sub-sampled from an existing dataset, this would leave a tiny number of data points that are insufficient to train a model. To this end, our experiments require access to a dataset's ground-truth renderer so latents can be sampled densely. This is not available in most cases, however. We now discuss this in a paragraph in App. B.6.
>
> ___
> **Comment:** “What if you … allow for a low-expressivity non-linear combination at the end for rendering?”
>
> **Response**:
> Regarding a “low-expressivity non-linear combination at the end for rendering”, we note that the softmax in the alpha-masking of Slot Attention in our experiments in Sec. 6.2 can be understood as just this. For these experiments, the model optimizes the consistency loss and includes a deterministic encoder, thus matching all theoretical assumptions except for _additivity_. In these cases, we found that the nonlinear combination leads to a steep increase in the isolated decoder OOD reconstruction error (See Fig. 5, left). Developing theory to allow for such nonlinear combinations is an important direction for modeling real-world data. We view our work as a crucial initial step in this direction.

---

> > ### Comment · Reviewer_JA2G · 2023-11-21
> > **Keeping my score**
> >
> > Thank you for your response! I appreciate the additions to the appendix, and they resolve my questions. I think this is a theoretically important paper, even though it is limited by its currently demonstrated applicability. I am going to keep my score.

---

### Official Review · Reviewer_wPGp · 2023-10-30

**Soundness:** 3 good
**Presentation:** 4 excellent
**Contribution:** 3 good
**Rating:** 8
**Confidence:** 3

**Summary:**

This paper presents conditions where compositional generalization is theoretically guaranteed for object-centric learning.

Specifically, they first extend the identifiability theory of object-centric representations to handle partial joint distribution supports, with an additional assumption/constraint on the decoder to be compositional. This ensures that slots are identifiable in the training distribution. They then ensure the generalizability of decoders (which, e.g., generate images given slot representations) with another assumption/constraint as the decoder being additive. The theoretical analysis is similar to those proving the compositional generalizability of any additive inference models.

The novel step is to enforce the compositional generalizability of encoders by learning with the synthesized data in new compositions of latent slots/symbols given the generalizable decoder. So, in order to learn an encoder that can generalize to unseen combinations of objects, they first build a dataset with new compositions of latent symbols/slots by permutating learned latent symbols/slots in the training distribution. They then generate the fake images using the "supposedly generalizable" decoders on new combinations. The encoder is trained to learn the inverse mapping of the decoder. This process is formulated as a compositional consistency regularization loss in practice.

The experimental results are aligned with the theories in a simple two-object synthetic image environment.

**Strengths:**

This paper discusses an important problem: learning compositionally generalizable object-centric representations. The paper is well-written and easy to read. The connections with related works are also interesting and inspiring.

The reviewer especially appreciates the theoretical guarantees and analysis. Even though the assumptions are strong on both the functions to be approximated as well as the parameterization of learned functions, they are still aligned with the image object-centric representation learning setting, and the methods can be relaxed and realized using modern object discovery methods such as slot attentions.

The proposed regularization loss to enforce the compositional generalizability of encoders is interesting and seems easy to use.

The ablation study on the additive decoder (softmax v.s. sigmoid in slot attentions) is interesting and inspiring.

**Weaknesses:**

It would be great if the assumptions could be relaxed, e.g., to handle occluded objects or to handle general latent variable learning domains other than the image objects.

The "contemporary" work [1] discussed most parts of this paper except for the generalizable encoder.

The experimental environment is simple with two-object synthetic images. It would be more convincing to see results on multi-object real images.

[1] S ́ ebastien Lachapelle, Divyat Mahajan, Ioannis Mitliagkas, and Simon Lacoste-Julien. Additive decoders for latent variables identification and cartesian-product extrapolation. arXiv preprint arXiv:2307.02598, 2023.

**Questions:**

Are there results in more complex environments?

Can the theories be generalized to more general settings with weaker assumptions?

---

> ### Author Response · Authors · 2023-11-21
>
> Dear Reviewer,
>
> We thank you for your positive review and your valuable feedback. We are happy to hear that you “appreciated” our theoretical analysis, that you found our work “well-written” and “easy to read”, and that you found multiple aspects of our work “interesting and inspiring”.
>
> We address your comments below:
>
> ___
> **Comment:** ”It would be great if the assumptions could be relaxed.”
>
> **Response**:
> We agree with the reviewer that relaxing our theoretical assumptions, namely _additivity_ and _compositionality_, is essential for modeling realistic data to allow for more complex interactions between slots. We aimed to highlight this in our discussion paragraph, “Extensions of Theory”.  While interactions between slots cannot be arbitrary, it is likely that further degrees of interaction may be allowed beyond additive interactions. We view this as an important direction for future work and believe that the current work constitutes a crucial step towards a more general result.
>
> ___
> **Comment:** "Are there results in more complex environments?"
>
> **Response**:
> Understanding our theoretical results for more complex data is indeed important for better understanding its applicability. This is challenging for real-world datasets, however, as one does not generally have access to the ground-truth latents making our evaluation schemes inapplicable. Moreover, using more complex synthetic datasets typically used in object-centric learning is also not straightforward for our experiments due to the need to sample latents densely from a slot-supported subset. Specifically, our experiments rely on sampling from a diagonal strip in the latent space with a small width. If such a region were sub-sampled from an existing dataset, this would leave only a tiny number of data points that are insufficient to train a model. To this end, our experiments rely on access to the ground-truth renderer for a dataset such that latents can be sampled densely; however, this is not available in most cases. We now discuss this point in a paragraph in App. B.6.

---

> > ### Comment · Reviewer_wPGp · 2023-11-22
> >
> > I appreciate the authors' reply and their honesty. I will then keep my rating as 8: accept, good paper.

---

### Official Review · Reviewer_KGuC · 2023-10-31

**Soundness:** 2 fair
**Presentation:** 3 good
**Contribution:** 2 fair
**Rating:** 6
**Confidence:** 3

**Summary:**

The paper addresses the problem of compositional generalization in object-centric autoencoders.
The authors formalize this as requiring the model to identify the ground-truth object latents not just on the training distribution, but also on out-of-distribution combinations.
They make two key assumptions to achieve this: (1) The generative process satisfies compositionality, meaning each pixel depends on one object, and irreducibility, preventing objects from being decomposed.
(2) The decoder is additive, decoding each object slot independently.
Under these assumptions, the authors prove autoencoders can identify objects in-distribution by minimizing reconstruction error.
The additive decoder then guarantees generalization out-of-distribution.
However, the encoder may still fail to generalize.
To address this, the authors propose compositional consistency regularization.
This trains the encoder to invert the decoder on recombined object slots, enabling the full autoencoder to generalize.
By combining in-distribution identifiability and compositional consistency regularization, the authors prove autoencoders satisfying their assumptions will generalize compositionally.
Through synthetic experiments, they provide empirical evidence supporting their theoretical results.
In particular, they demonstrate the importance of additivity and compositional consistency for generalization.

**Strengths:**

This paper made contributions for
- Formalizing compositional generalization as an identifiability problem
- Theoretical guarantees for in-distribution identifiability
- Showing an additive decoder enables out-of-distribution generalization
- Introducing compositional consistency regularization
- Providing overall theoretical guarantees for compositional generalization

The work makes theoretical progress on understanding compositional generalization in object-centric representation learning.

**Weaknesses:**

- The assumptions of compositionality and irreducibility are quite restrictive. Most real-world datasets likely violate these.
- The additive decoder limits modeling of complex object interactions and relations.
- The consistency regularization implementation requires sampling implausible object combinations. More principled schemes could improve results in complex environments.
- Experiments only validate the theory on simple synthetic datasets. Testing on more diverse and realistic data would better demonstrate applicability, though the evaluation would also be more challenging.
- The proposed methods, especially when ensuring encoder-decoder consistency and handling latent slots, might pose scalability issues for very large datasets or more complex models. A discussion on the scalability, computational costs, and potential solutions would make the paper more robust.

**Questions:**

Please see above.

---

> ### Author Response · Authors · 2023-11-20
>
> Dear Reviewer,
>
> We thank you for your review and your valuable feedback. We are happy that you found our work makes “theoretical progress on understanding compositional generalization in object-centric representation learning“.
>
> We address your comments below:
>
> **Comment:** “Compositionality and irreducibility are quite restrictive.”
>
> **Response**:
>
> Regarding compositionality, we agree with the reviewer about the restrictiveness of this assumption. Compositional decoders are special cases of additive decoders (see App. A.7) and thus share the same limitations: They cannot perfectly model objects with complex interactions such as occlusion or reflection. Irreducibility, on the other hand, roughly states that parts of the same object share information. We argue that this assumption is a more fundamental property of objects and will thus be valid more generally for multi-object scenes.
>
> **Comment:** “Additive decoder limits modeling of complex object interactions and relations.”
>
> **Response**:
>
> As noted in our “Extensions of Theory” paragraph in the Discussion section, we agree with the reviewer that additivity is indeed insufficient for modeling more complex object interactions. We believe, however, that this assumption can be relaxed to allow for more complex interactions and view this as an important direction for future work.
>
> **Comment:** ”Consistency regularization requires sampling implausible object combinations. More principled schemes could improve results.”
>
> **Response**:
>
> As noted in our “Extensions of Experiments” paragraph in the Discussion section, we agree regarding the limitations of our consistency loss implementation and approaches for improving it. One such approach could be to learn a prior over latent slots and sample slot combinations more rigorously according to their likelihood under the prior. This could avoid sampling implausible combinations; however, such a scheme is challenging as it relies on the likelihood of being valid for OOD combinations of slots. Another possibility would be to include heuristics to directly filter combinations based on prior knowledge of the data-generation process, e.g., to filter objects with similar coordinates that would intersect. We believe that further explorations in such directions are a promising and important direction for future work to make this method practical for more complex datasets. We now include this discussion in App. B.5.
>
> **Comment:** “Experiments only validate the theory on simple synthetic datasets.”
>
> **Response**:
>
> We agree with the reviewer that understanding our theoretical results on more diverse and real-world data is important for better understanding its applicability. This is indeed challenging for real-world datasets, however, as one does not generally have access to the ground-truth latents making our evaluation schemes inapplicable. Moreover, using the synthetic datasets typically used in object-centric learning is also not straightforward for our experiments due to the need to sample latents densely from a slot-supported subset. Specifically, our experiments rely on sampling from a diagonal strip in the latent space with a small width. If such a region were sub-sampled from an existing dataset, this would leave only a tiny number of data points that are insufficient to train a model. To this end, our experiments rely on access to the ground-truth renderer for a dataset such that latents can be sampled densely; however, this is not available in most cases. We now discuss this paragraph in App. B.6.
>
> **Comment:** “The proposed methods might pose scalability issues for very large datasets or more complex models.”
>
> **Response**:
>
> Thank you for mentioning this. Methods based on latent slots have in fact been shown to scale to more complex settings (e.g., [1, 2]); however, we agree regarding the scalability of the consistency loss, which we aimed to highlight in our “limitations of experiments” paragraph. We have included a figure in App. C.3 highlighting the challenges in scaling this loss as the number of objects increases. Regarding computational costs, the loss requires additional passes through the encoder and decoder as well as computation of the encoder’s gradients wrt. the loss. We found this to increase training time by a maximum of 28% across runs. We have incorporated a discussion of this point in App. B.5.
>
> We hope these clarifications and revisions are sufficient for the Reviewer to confidently increase their score. Furthermore, we noticed the Reviewer left a score of “2” regarding the soundness of our manuscript. We found this noteworthy as we are not aware of any inaccuracies in our theoretical or empirical contribution. Thus, if the soundness score was the result of one of the points raised by the reviewer above, we hope our comments were sufficient to resolve the reviewer's doubts.
>
> [1] https://arxiv.org/abs/2206.07764
>
> [2] https://arxiv.org/abs/2205.14065

---

### Author Response · Authors · 2023-11-21

We thank all reviewers for their time and positive assessment of our work. Namely, the assessment of our theory as “sound and significant for the community” (JA2G) and “making progress” (KGuC), aspects of our experiments as “interesting and inspiring” (wPGp), and our manuscript as “very well-written” (JA2G) and “easy to read” (wPGp).

We also appreciate the constructive feedback from all reviewers. Based on your comments, we have added the following content to the paper:

- App. B.5 discusses the scalability and computational cost of the proposed consistency loss.
- App. B.6 discusses the challenges of verifying our theory on more realistic datasets.
- App. C contains additional experiments and figures:
    - App. C.1 shows the effect of the size of the slot-supported subset for models trained with and without the consistency loss
    - App. C.2 illustrates the importance that the latent space is a slot-supported subset
    - App. C.3 shows how the consistency loss scales as the number of slots is increased

---

### Comment · Area_Chair_4jrk · 2023-11-23
**Discussion between authors and reviewers**

Dear Reviewers,

Thanks for the reviews. The authors have uploaded their responses to your comments, please check if the rebuttal address your concerns and if you have further questions/comments to discuss with the authors. If the authors have addressed your concerns, please adjust your rating accordingly or vice versa.

AC

---

### Meta-Review · Area_Chair_4jrk · 2023-12-03

**Metareview:**

This paper investigates the sufficient conditions for compostional generalization(CG) of Deep Neural Networks. By formalizing compositional generalization as an identifiability problem, it proves theoretically that the object-centric autoencoders with structural models and consistent encoder-decoder can learn features generalizing compositionally. Experiments on synthetic data validate the theoritical results.

Strengths:
+ The paper addresses an important problem on ML.
+ The theory proposed in this paper on CG is technically sound.
+ The proposed compositional consistency regularization helps achieve CG.
+ The experiments validate the theoretical results.
+ The paper is well-written and easy to read.

Weaknesses:
- The assumptions of compositionality and irreducibility are very restrictive and might not applicable to some real-world datasets.
- The experiments are not conducted on real-world data but synthetic ones.
- Computation of consistency loss might be expensive and not scalable to real-world problems.

**Justification For Why Not Higher Score:**

N/A

**Justification For Why Not Lower Score:**

The paper addresses an important problem on ML, and it makes theoretical progress on understanding compositional generalization in object-centric representation learning. The theory proposed in this paper on CG is technically sound and significant to the ML community.

---

### Decision · Program_Chairs · 2024-01-16

Accept (oral)